

# HYPHOP: a tool for high-altitude, long-range monitoring of hydrogen peroxide and higher organic peroxides in the atmosphere

Zaneta Hamryszczak[1], Antonia Hartmann[1], Dirk Dienhart[1], Sascha Hafermann[1], Bettina Brendel[1], Rainer Königstedt[1], Uwe Parchatka[1], Jos Lelieveld[1,2] and Horst Fischer[1]

[1]Atmospheric Chemistry Department, Max Planck Institute for Chemistry, Mainz, 55128, Germany
[2]Climate and Atmosphere Research Center, The Cyprus Institute, Nicosia, 1645, Cyprus

*Correspondence to*: Zaneta Hamryszczak (z.hamryszczak@mpic.de) and Horst Fischer (horst.fischer@mpic.de)

**Abstract.**

Measurements of hydroperoxides help improve our understanding of atmospheric oxidation processes. Here, we introduce an instrument setup designed for airborne hydroperoxide measurements. The HYdrogen Peroxide and Higher Organic Peroxides (HYPHOP) monitor has been deployed on the German High-Altitude and Long-range Observatory (HALO) aircraft and is based on dual-enzyme fluorescence spectroscopy, enabling measurements up to ambient pressure of approximately 150 hPa pressure altitude (13.5–14 km). We characterized the measurement method and data acquisition of HYPHOP with special emphasis on potential sources of interference impacting instrument uncertainty. Physically derived interference was examined based on a dedicated test flight to investigate potential measurement inconsistencies arising from the dynamic movement patterns of the aircraft. During the test flight, the hydroperoxide monitor was operated in the background air sampling mode with purified air by scrubbing atmospheric trace gases, to investigate the instrument stability and potential parameters that might affect the measurements. We show that technical and physical challenges during flight maneuvers do not critically impact the instrument performance and the absolute measurements of hydroperoxide levels. Dynamic processes such as convective transport in the South Atlantic Convergence Zone (SACZ) are well-resolved as shown in the overview of a recent measurement campaign, Chemistry of the Atmosphere: Field Experiment in Brazil, in December 2022–January 2023 (CAFE-Brazil). The instrument precision based on the measurement results during CAFE-Brazil for hydrogen peroxide and the sum of organic hydroperoxides is estimated to be 6.4% (at 5.7 ppbv) and 3.6% (at 5.8 ppbv), respectively, and the corresponding detection limits 20 pptv and 19 pptv for a data acquisition frequency of 1 Hz, subsequently integrated over 120 second time intervals.

## 1 Introduction

Hydroperoxides are key contributors to the self-cleaning capacity of the troposphere due to their dual role as sinks and sources of the main atmospheric oxidant, the hydroperoxyl radical (OH) and of peroxy radicals (HO$_2$), often collectively described as HO$_x$ (HO$_x$= HO$_2$+OH; Gunz and Hoffmann, 1990; Lee et al., 2000; Reeves and Penkett, 2003 and the references therein). Due to their relatively high solubility and reactivity, hydroperoxides, especially H$_2$O$_2$, play an important role in the chemistry of




the liquid phase of clouds, rain, and fog (Kelly et al., 1985; Madronich, 1987; Olszyna et al., 1988; Sakugawa et al., 1990; Lelieveld and Crutzen, 1991, 1994; Edy et al., 1996).

Schöne et al. reported the first findings on the abundance of hydrogen peroxide ($H_2O_2$) in the atmospheric aqueous phase in the 19[th] century (1874, 1893, 1894), but it was only in the 1970s that hydroperoxides became the focus of significant scientific
attention (Gunz and Hoffmann, 1990 and references therein). The 1980s saw extensive work on the role of hydroperoxides in the generation of sulfuric acid and nitric acid in clouds, rain, and fog leading to characterization of the phenomena of so-called acidic fog and rain (Hoffmann and Edwards, 1975; Penkett et al., 1979; Robbin Martin and Damschen, 1981; Damschen and Martin, 1983; Kunen et al., 1983; Calvert et al., 1985; Lee and Lind, 1986). More recent studies on the hydroperoxides have provided insights into their tropospheric abundance and their importance in atmospheric oxidative processes (Gunz and
Hoffmann, 1990; Sakugawa et al., 1990; Reeves and Penkett, 2003; Klippel et al., 2011; Fischer et al., 2019; Hottmann et al., 2020; Allen et al., 2022a; Allen et al., 2022b; Hamryszczak et al., 2022; Dienhart et al., 2023; Hamryszczak et al., 2023). Thus, it is not surprising that over the past decades, numerous hydroperoxide measurement methods have been established to investigate the species in both the gaseous and aqueous phases of the atmosphere.

The first highly sensitive $H_2O_2$ measurements were performed using luminol and peroxy oxalate-based chemiluminescent
techniques (Kok et al., 1978; Bufalini et al., 1979; Kelly et al., 1979; Kok, 1980; Heikes et al., 1982; Römer et al., 1985; Heikes et al., 1987; Jacob and Klockow, 1992). Guilbault et al. (1968) proposed an enzyme fluorescence technique using peroxidase to detect hydrogen peroxide Later, Lee et al. (1990; 1994) presented an alternative non-enzymatic measurement technique based on the $H_2O_2$ Fenton reaction. Alternative methods, such as high-performance liquid chromatography (HPLC) paired with post-column derivatization (Hellpointner and Gäb, 1989; Hewitt and Kok, 1991; Fels and Junkermann, 1994; Kok et al.,
1995; Lee, 1995) and tunable diode laser absorption spectroscopy (TDLAS; Slemr et al., 1986; Mackay et al., 1990) have likewise been widely adopted for measuring $H_2O_2$ in ambient air. Further, gas chromatography techniques were used to sample hydrogen peroxide and ROOH organic hydroperoxides (Kok et al., 1995 and the references therein). The most recently developed and optimized technique described by Crounse et al. (2006) and St.Clair et al. (2010) uses measurements based on soft chemical ionization and compact time-of-flight mass spectroscopy complimented by triple quadrupole mass spectrometry.
The method was used to detect hydroperoxides in the troposphere with high sensitivity (California Institute of Technology Chemical Ionization Mass Spectrometers, CIT-CIMS).

The measurement technique presented in this work is based on the wet chemical dual-enzyme measurement system of Lazrus et al. (1985; 1986), which has been successfully employed in numerous research groups (Gunz and Hoffmann, 1990 and references therein). With this method, hydroperoxides are detected using p-hydroxyphenyl acetic acid (POPHA) and
horseradish peroxidase (HRP). The stoichiometric reaction of these compounds yields a hydroperoxide-specific chemiluminescent molecule (Guilbault et al., 1968), which is detected by fluorescence spectroscopy. The distinction between $H_2O_2$ and ROOH is achieved by selective destruction of the former with catalase in a dual-reactor system.

To meet the requirements of long measurement flights under highly dynamic airborne conditions, the Hydrogen Peroxide and Higher Organic Peroxides (HYPHOP) instrument was developed based on the commercially available AL2001CA





hydroperoxide monitor (Aero-Laser, Garmisch-Partenkirchen, Germany). HYPHOP was designed as a lightweight and compact monitor, where the measurement process and the complementary data acquisition are controlled by a compact V25 control unit (Max Planck Institute for Chemistry, Mainz, Germany). The instrument was further equipped with a constant pressure inlet unit (CPI), making it suitable for *in situ* measurements under varying ambient air pressures. HYPHOP was designed in 2014 and was first installed onboard of the High Altitude and Long-range research aircraft (HALO) in January

2015 during the Oxidation Mechanism Observation (OMO) project. Since that time, it has been deployed successfully in 53 measurement flights with more than 330 flight hours of tropospheric and lower stratospheric hydroperoxide measurements during aircraft campaigns (Hottmann et al., 2020; Hamryszczak et al., 2022; Hamryszczak et al., 2023). The instrument is deployed together with the Tracer In-Situ TDLAS (Tunable Diode Laser Absorption Spectroscopy) for Atmospheric Research (TRISTAR) in a 19'' measurement rack and uses two measurement channels for continuous measurements (Tomsche et al.,

2019). HYPHOP tracks total levels of ambient hydroperoxides and the sum of organic peroxides separately. By subtracting the organic hydroperoxide mixing ratios from the total measured hydroperoxides, hydrogen peroxide mixing ratios are determined.

Based on studies of hydroperoxide reactions with other tropospheric trace gases, different chemical sources of interference affecting the measurement method have been identified (Heikes et al., 1982; Graedel and Goldberg, 1983; Graedel et al., 1986;

Weschler et al., 1986; Jackson and Hewitt, 1999). On the one hand, reactions with sulfur dioxide, ($SO_2$), metal ions and nitrogen oxide (NO) in the atmospheric aqueous phase lead to a significant loss of the ambient $H_2O_2$ in liquid samples. On the other hand, $H_2O_2$ production via adsorption of ozone ($O_3$) on wet surfaces leads to positive interference in the measurement system. Additionally, due to the specific measurement setup, instrument precision may suffer from various physical noise sources, such as cabin temperature altering the solubility, mainly of organic hydroperoxides, in the analytic solution and incomplete

transmission of the trace gases due to wall loses in the inlet of the system. Detection instabilities might also be triggered by rapid aircraft movements and turbulence affecting both the liquid and gaseous mass transport during the operation.

This work describes the measurement method and characteristics and addresses potential sources of inconsistencies caused by chemical and physical interference during measurement flights. To examine the effect of physical interference on *in situ* hydroperoxide measurements, an individual test flight to examine the impact of aircraft movement patterns on the measurement

outcome was designed and performed during the preliminary phase of the Chemistry of the Atmosphere – Field Experiment in Brazil (CAFE-Brazil; October–November 2022) campaign based out of the German Aerospace Center (DLR) in Oberpfaffenhofen (Germany). A detailed introduction to the instrument setup on board of the research aircraft, the measurement method, data acquisition, and signal correction are given in the following sections (Sect. 2.1 and Sect. 2.2). Additionally, we address potential chemically and physically derived inconsistencies, analyze their impact on airborne

hydroperoxide measurements, and characterize the instrument in detail (Sect. 2.3 and 2.4). In Sect. 3, an overview of the instrument performance is given based on the measurement results during the CAFE-Brazil aircraft campaign. Finally, we present an overview of the characteristics of the hydroperoxide monitor (Sect. 4).



## 2 HYPHOP: HYdrogen Peroxide and Higher Organic Peroxide monitor

### 2.1 Measurement method and instrument setup on board of the HALO research aircraft

During airborne measurements performed with the research aircraft HALO, the HYPHOP monitor is deployed into a single
19" measurement frame rack together with the TRISTAR instrument. The rack is further equipped with a manually cooled
liquid container department and a constant pressure inlet (CPI) pump. Both HYPHOP and TRISTAR are simultaneously
connected to a bypass that channels the sampled ambient air from the Teflon-coated trace gas inlet (TGI) embedded in a
stainless-steel compartment. Fig. S1a provides a picture of the instrumental setup on board of HALO. Since the commercially
available peroxide monitor was designed to perform measurements under standard conditions, the airflow system of HYPHOP
was extended by the CPI system, ensuring constant pressure of the sampled air during airborne measurements at highly variable
ambient pressure conditions. The technical circuit of the system is presented as a diagram below (Fig. 1).

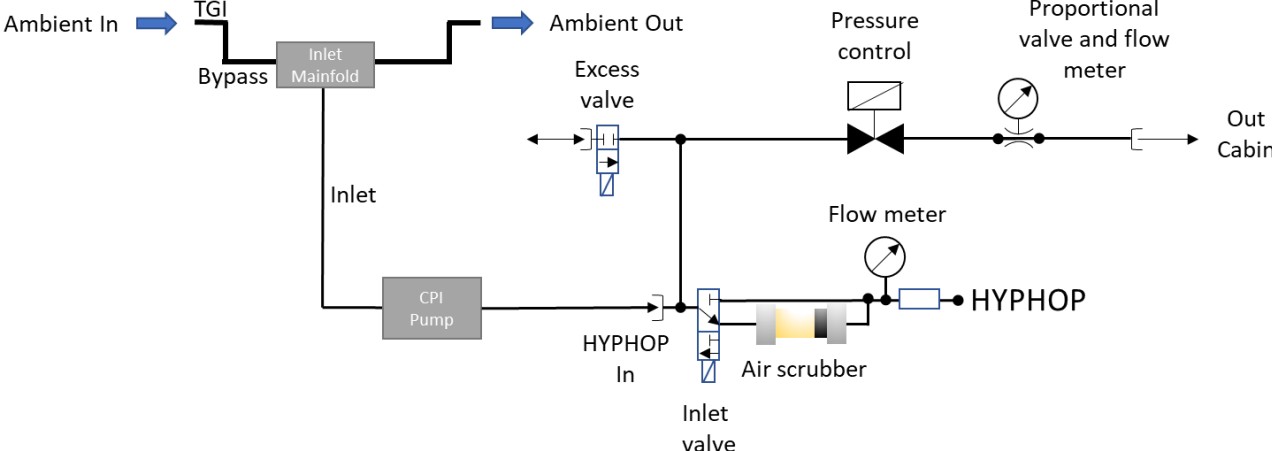

**Figure 1: Technical circuit diagram of the airflow system.**

During airborne measurements, ambient air is sampled from the top of the aircraft fuselage through a Teflon-coated tubing
installed in a forward-facing stainless-steel TGI connected to a bypass consisting of a 1.45 m-long ½ in. PFA (perfluoro alkoxy
alkane) tube inside the aircraft and to an exit through a second TGI. The gas inlet system consisting of connection tubing from
the bypass via a Teflon inlet manifold (length: 0.27 m) to the CPI pump (length: 2.65 m) and further to the monitor (length:
1.15 m) is made of a ¼ in. PFA tubing to reduce potential surface effects and wall losses. The CPI system contains an internal
CPI control unit and a Teflon-coated membrane pump (type MD 1C; Vacuubrand, Wertheim, Germany), generating an excess
airflow of up to approximately 10 standard liters per minute (slm) to the sampling inlet of the monitor. The CPI is connected
via the inlet manifold to an external pressure sensor tracking the ambient pressure (inlet pressure). The internal CPI control
unit consists of a proportionality valve with an airflow sensor, an excess valve, and a pressure control unit in front of the
sampling coil to ensure nearly constant pressure during the measurement flights. The highly sensitive proportionality valve





controls the airflow through the instrument based on variations of the measured ambient pressure detected by the sensor in the inlet, so that a stable airflow of approximately 2 slm is achieved. Excess air is ejected into the cabin. The additional excess valve is activated when the measured air pressure of the sampling line exceeds 1100 hPa, thus preventing damage to the instrument due to overpressure. With higher altitudes, the pump output increases, and, thus, the line pressure and the airflow

are stabilized to approximately ambient pressure and 2 slm.

After traversing the inlet system, the ambient air is passed through a reaction coil (glass) with a simultaneous injection of a precooled buffered sampling solution (4–6 °C; pH = 5.8–6.0) consisting of potassium hydrogen phthalate (KHP) and sodium hydroxide (NaOH; 1 M) in purified water. From the reaction coil, the hydroperoxide-enriched sampling solution is subdivided into two reactors (channels A and B), where an analogously precooled fluorescence solution consisting of p-hydroxyphenyl

acetic acid (POPHA) and horseradish peroxidase (HRP) and a precooled sodium hydroxide solution (NaOH; 1 M) are added. Fig. 2 gives an overview of the liquid flow circuit of HYPHOP. The front view of the instrument is displayed in Fig. S1b in the Supplement of this work.

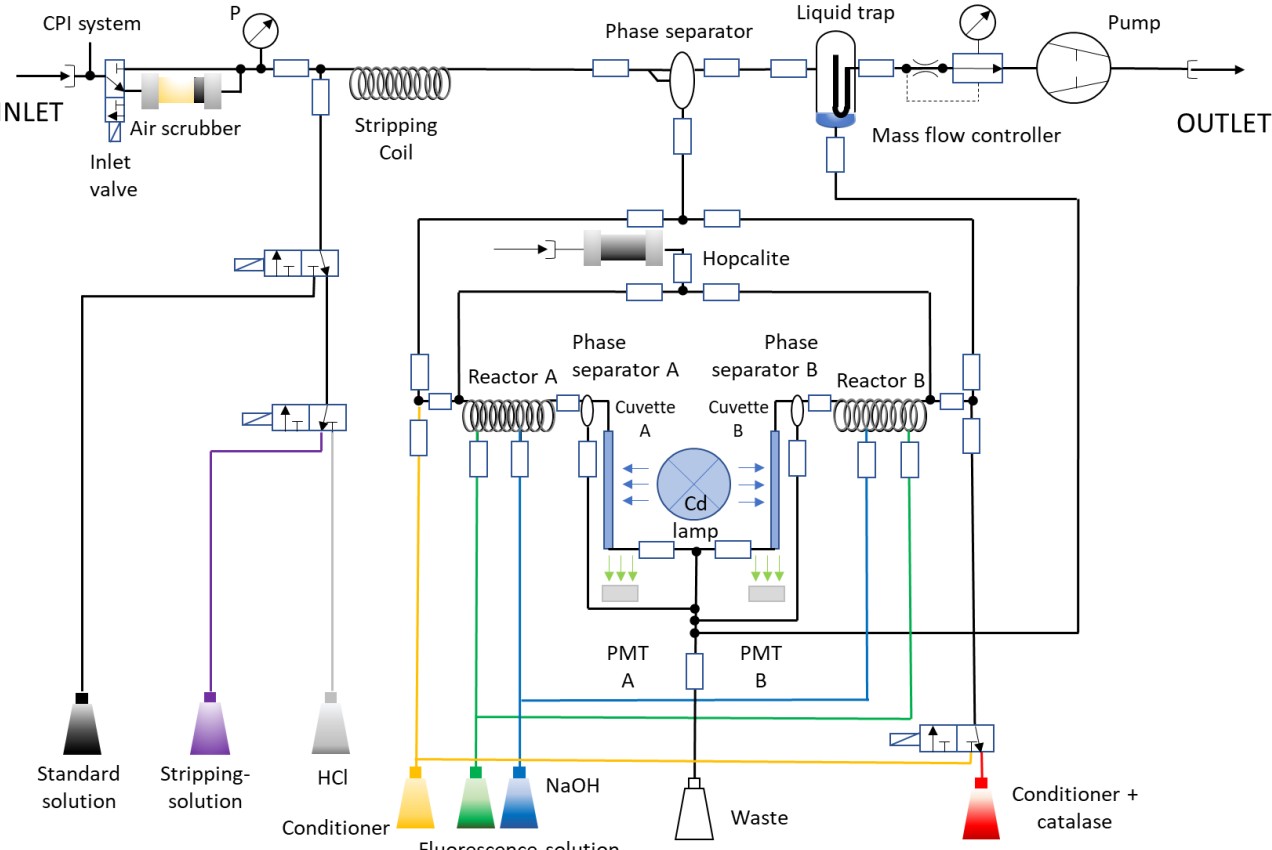

**Figure 2: Liquid flow diagram of HYPHOP.**



The hydroperoxides are detected based on stoichiometric reaction with POPHA, which yields a chemiluminescent POPHA-
dimer (6,6'-dihydroxy-3,3'-biphenyl diacetic acid). The catalytic center of HRP enables a species-specific binding of the
hydroperoxides, which subsequently react with POPHA to form chemiluminescent-active dimers. This chemiluminescent
compound is then measured with a Cd pen ray lamp (UVP, Inc., Upland, United States) at 326 nm by means of fluorescent
spectroscopy. The species-specific fluorescence is detected at 400–420 nm using two parallel photomultiplier tubes (PMT;
Type: H957-29, Hamamatsu, Japan) at both channels separately as electrical signals, which are translated into mixing ratios
using a four-point calibration described in Sect. 2.2. The concentration of $H_2O_2$ in the sample is determined as the difference
between the total concentration of hydroperoxides (channel A) and the sum of the organic peroxides (ROOH; channel B),
where the added catalase selectively destroys hydrogen peroxide in the sampling solution. The corresponding chemical
reactions (SR1–SR4) are presented in the Supplement of this work. The catalase destruction efficiency of hydrogen peroxide
is determined via liquid calibration of the instrument at 0.95–0.99 using an $H_2O_2$ liquid standard (0.99 µmol $L^{-1}$) produced in
a serial dilution from a constantly cooled $H_2O_2$ stock solution.

## 2.2 Data acquisition and signal correction

HYPHOP operates in three modes: liquid calibration, background and ambient mode. The monitor is controlled by a V25
control unit (Max Planck Institute for Chemistry, Mainz, Germany). The control unit is a multitasking, multiprocessing real-
time operating system, which consists of a command interpreter and a configuration compiler for mode application, data
storage, and PC communication. The valves switch between the supply tubing of the solutions required for each mode and
between ambient and purified air, respectively, depending on the operating mode which is managed by the V25 unit (Fig. 2).
During the ambient (measure) mode, the sampling process is performed as described above. In the background mode,
hydroperoxide-free air samples are produced and measured frequently in order to investigate any temperature-related altering
of the hydroperoxide signals during the flights. The background air is generated by transmitting ambient air through a scrubber,
the zero-gas cartridge, containing silica gel (type IAC-502; Infiltec, Speyer, Germany), which adsorbs the moisture from the
air sample and hopcalite (type IAC-330; Infiltec, Speyer, Germany). The latter destroys atmospheric hydroperoxides and other
trace gases, enabling estimation of the instrument background, and is used during the calibration.

The calibration mode is performed on the ground prior to each measurement flight, during which a liquid $H_2O_2$ standard is
measured in order to determine the hydroperoxide detection sensitivity and to estimate a signal reference required to transfer
the detected electric signal into hydroperoxide mixing ratios. During this process, the instrument generates constantly purified
hydroperoxide-free air analogous to the background measurement procedure. Prior to the calibration, both channels are flushed
with hydrochloric acid (HCl; 1 M) in order to remove remaining catalase and any potential residuals from previous ambient
measurements. The calibration process is subdivided into four phases (four-point calibration) using a freshly prepared liquid
standard (2–4 °C) obtained from a serial dilution of a constantly cooled hydrogen peroxide stock solution. In the first step of
the liquid calibration, the standard is injected into the sampling coil. In the second and third steps, purified air is sampled and



channel B is operated with and without the addition of catalase. In the last step, the liquid $H_2O_2$ standard is injected again with the addition of catalase in channel B.

The concentration of hydrogen peroxide ($[H_2O_2]$) in the liquid standard is determined by means of a redox titration with

potassium permanganate ($KMnO_4$) under acidic conditions ($H_2SO_4$) as presented in the Supplement of this work (SR5–SR7 and Eq. S1). The corresponding mixing ratios can be calculated using the measured air and liquid flows of the instrument. Based on the four-point calibration process, the sensitivities of the channels ($s_A$, $s_B$) are determined (Eq. 1–2).

$$s_A = \frac{U_{A,S} - U_{A,0}}{\mu_S} \tag{1}$$

$$s_B = \frac{U_{B,S} - U_{B,0}}{\mu_S} \tag{2}$$

Here, $U_{A,S}$ and $U_{B,S}$ are the measured voltages during the first phase of the liquid calibration in the respective channels, and $U_{A,0}$ and $U_{B,0}$ are the corresponding signals measured in the second phase of the calibration. $\mu_S$ is the mixing ratio of the standard solution under consideration of the molar air volume ($V_m$) and the measured standardized mass flows ($Q_{Stripping}$, $Q_{Air}$; Eq. S2 – S3):

$$\mu_S = \frac{[H_2O_2]}{10^4} \cdot 10^9 \cdot V_m \cdot \frac{Q_{Stripping}}{Q_{Air}} \tag{3}$$

The corresponding catalase destruction efficiency of hydrogen peroxide in channel B, $\varepsilon$, is derived from the following equation:

$$\varepsilon = 1 - \left( \frac{U_{BK,S} - U_{BK,0}}{U_{B,S} - U_{B,0}} \cdot \frac{U_{A,S} - U_{A,0}}{U_{AK,S} - U_{AK,0}} \right) \tag{4}$$

with the voltages (U) in both channels during the consecutive phases of the liquid calibration (liquid standard injection: S; background measurement: 0; background measurement with catalase: K,0; liquid standard injection with catalase: K, S).

Based on the specific sensitivities, and the destruction efficiency, the absolute peroxide mixing ratios $\mu_{H_2O_2}$ and $\mu_{ROOH}$ in the sample are determined using the following equations:

$$\mu_{H_2O_2} = \frac{U_A - U_{AK,0}}{s_A} - \frac{(U_B - U_{BK,0})}{s_B} \cdot \varepsilon \tag{5}$$

$$\mu_{ROOH} = \frac{U_B - U_{BK,0}}{s_B} - (1 - \varepsilon) \cdot \mu_{H_2O_2} \tag{6}$$

Due to the characteristics of the wet chemical measurement method, signal corrections have to be initially performed in order to obtain absolute hydroperoxide mixing ratios. In order to account for potential measurement divergencies and background signal alterations initiated by pressure and temperature instabilities during the measurement flights, the frequently measured background signal is interpolated. Further, the tracked and interpolated background signal is subtracted from the measurements to give the actual hydroperoxide signals in the ambient air.

The time delay caused by the liquid transport process of the monitor is accounted for by shifting the obtained signals by the time difference between the valve switches and the corresponding signal response during the background measurements. The



time modification is calculated as the mean time delay based on all performed background measurements during the respective flight.

To account for wall loses at the inner surface of the bypass, the inlet, and the Teflon-coated CPI pump, the inlet transmission
efficiency has to be determined. Due to the high flow rates through the installed bypass and attached inlet (30 slm and 10 slm, respectively), potential wall losses in the tubing are assumed to have a minor impact on the hydroperoxide measurements. The Teflon coating of the membrane pump is expected to absorb hydroperoxides, and, thus, decrease the species inlet transmission from the TGI to the monitor. The subsequent wall loss at the inner surface of the CPI pump is measured using hydrogen peroxide and peroxy acetic acid (PAA) gas standards produced by permeation sources. The sources are operated at 28 °C and
40 °C based on a $H_2O_2$ (30%) or PAA solution (37%), respectively, which is filled into a 1/8" LDPE tubing with an approximate length between 8–20 cm, depending on the desired mixing ratio. The permeation device is flushed with purified air at approximately 60 standard cubic centimeters per minute (sccm) and subsequently diluted with approximately 10 slm of purified air. The purified air is generated by the removal of trace gases from ambient air using zero gas cartridges containing silica gel and hopcalite as described above. The generated gas standard is injected into the inlet with and without the CPI pump in the
air sampling system. The inlet transmission efficiency is then determined based on the detected hydroperoxide mixing ratios with the addition of the CPI pump relative to the detected mixing ratios without the CPI pump in the inlet system separately for $H_2O_2$ and ROOH. The permeation rate of the corresponding species is determined based on the work by Pilz and Johann (1974) by transmitting the permeation gas through three impinger flasks. Two of these are installed in series and a third flask serves to determine the chemical background. Spectrophotometrical determination of $H_2O_2$ in the collection samples is
achieved by the addition of acidified (HCl) titanium chloride solution and subsequent measurement of the absorption at 415 nm.

Due to the temperature-dependent solubility of hydroperoxides determined by Henry's law, the measurement results must be additionally modified relative to the efficiency of the sampling solution used to scrub the trace gases from the ambient air. The sampling efficiency gives the relative amount of the hydroperoxides transferred into the sampling solution and has been
investigated by numerous research groups in the past years (Lee et al., 2000 and references therein). Based on the precooled sampling solution (4–6 °C), we assume sampling efficiencies of 1 and 0.6 for $H_2O_2$ and the ROOH. Please note that the organic hydroperoxide sampling efficiency is based on the approximation that in the free troposphere methyl hydroperoxide (MHP) is the most prominent ROOH (Lee et al., 2000). However, the composition of the organic hydroperoxides is expected to vary in the boundary layer, i.e., in pristine regions with extended amounts of vegetation-related emissions or due to specific primary
emissions such as biomass burning events (Fels and Junkermann, 1994; Lee et al., 1997; Valverde-Canossa et al., 2005; Hua et al., 2008). MHP is known to be less soluble than, e.g., PAA and hydroxymethyl hydroperoxide (HMHP) in aqueous solutions, while ethyl hydroperoxide (EHP) is expected to be similarly soluble (O'Sullivan et al., 1996). Thus, the sampling efficiency of 0.6 is an estimate for the lower limit of the ROOH sampling efficiency. The transmission efficiency during the most recent field campaign was calculated to be 82% (± 1.6%) for $H_2O_2$ and 95% (±1.1%) for ROOH.



## 2.3 Potential sources of error: chemical and physical interferences

Based on numerous studies on the reactions of hydrogen peroxide with other tropospheric trace gases, the monitor's precision is negatively impacted by chemical interference induced by reactions with sulfur dioxide ($SO_2$), nitrogen oxide (NO), and metal ions in the liquid phase. Studies on acid rain have shown that the oxidation of $SO_2$ by $H_2O_2$ in cloud droplets might lead to a significant loss of the ambient hydrogen peroxide in the generated samples. Graedel and Goldberg (1983) described the reaction of hydrogen peroxide with iron ions ($Fe^{2+}$) analogous to the Fenton reaction. Comparable results were later found for organic hydroperoxides (Weschler et al., 1986). In 1982, Staehelin and Hoigne investigated the production of $H_2O_2$ by passing ozone through an impinger flask containing purified water. Heikes (1982; 1984) described production of $H_2O_2$ via adsorption of ozone on wet surfaces. The corresponding reactions are included in the Supplement of this work (SR8–SR16).

In order to account for the negative impact of reactions with metal ions and $SO_2$ on measurements, sufficient amounts of ethylene diamine tetra acetic acid (EDTA; 100 mg per 5 L), and of formaldehyde (HCHO; 1 mL per 5 L) are added to the buffered sampling solution. The corresponding reactions are presented in the Supplement (SR17–SR18). Further, the obtained data is corrected for the positive ozone interference by subtracting the determined amount of hydrogen peroxide per 100 ppbv of ozone. The correction factor is determined by plotting the measured $H_2O_2$ mixing ratios versus the $O_3$ mixing ratios measured under laboratory conditions or during measurement flights in the lower stratosphere under the assumption that ambient levels of hydrogen peroxide are close to zero above the tropopause due to the reduced availability of the $H_2O_2$ precursor $HO_2$. Based on the most recent measurements, the correction factor comes to 0.01 ppbv $H_2O_2$/100 ppbv $O_3$. The commercially available Aero-Laser instrument estimates a potential negative interference with NO in ambient air of 0.012 ppbv $H_2O_2$/100 ppbv NO, which is only relevant under highly polluted conditions.

Furthermore, physical errors might arise from interference between aircraft movement and the individual components of the monitor. Large ambient pressure variations might affect the airflow during the measurements. Dynamic pitch angle alterations, high descent or ascent rates of the aircraft, as well as instrument valves switching at maximum altitudes, are all factors that can lead to temporal sampling line pressure inconsistencies. Also, liquid transport might be affected by changes in the aircraft movement pattern, especially by pitch and roll angle changes during spiraling maneuvers. Furthermore, cabin temperature but also cabin pressure may have a significant effect on the signal detection in both channels.

The instrument performance and the impact of physical noise sources were examined during an individual test flight as part of the most recent CAFE-Brazil campaign performed on November 22, 2022 from the German Aerospace Center (Deutsches Zentrum für Luft- und Raumfahrt, DLR) base of operation in Oberpfaffenhofen (Germany; 48° 4' N, 11° 16' E). The four-hour flight consisted of numerous maneuvers including rapid flight direction changes and flight altitude variations at alternating ascent and descent rates (1,500–3,000 ft/min). The vertical maneuvers covered an altitude range of a few tens of meters up to 15,000 m above the earth's surface. The average aircraft speed was determined to be approximately $193 \pm 48.5$ m · s$^{-1}$. Fig. S2 shows the three-dimensional flight pattern of the aircraft during the test flight. During the flight, HYPHOP was operated in a



continuous background mode, sampling dried and peroxide-free air at approximately 2.3 slm and 4–6°C liquid solution temperature.

Fluctuations of the temperature reported during the flights (flight log book) might affect the measured temperature at the stripping coil. However, due to the precooled measurement solutions transported via the sampling coil, we assume the impact of any low-range temperature variability to be of minor importance for the solubility of hydroperoxides. Nonetheless, fluctuations in the cabin temperature might influence both the excitation and the detection process. Fig. 3 displays the temporal series of the tracked sampling coil temperature, the lamp voltage, and the PMT temperature with respect to the altitude.

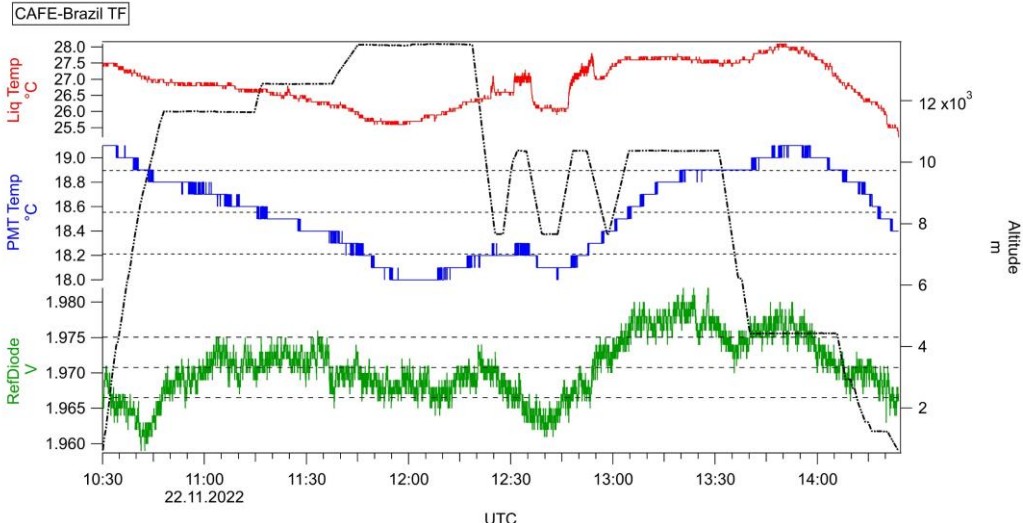

**Figure 3: Temporal series of the Cd lamp (reference diode; green) with the respective average and 1σ standard deviation range (dashed lines) during the test flight performed on November 22, 2022 complimented by the GPS flight altitude (right plot; black), observed temperature at the sampling coil (Liq Temp; red) and PMT temperature (blue). The data is displayed with 1-Hz measurement frequency.**

Temperature modifications within the cabin indicated by LiqTemp (red data) are clearly followed by inconsistencies in the Cd

lamp and the PMT signals. The calculated average of the lamp signal and the PMT temperature are $1.971 \pm 0.004$ V and $18.55 \pm 0.341$ °C, respectively. Nonetheless, as displayed in Fig. 3 (bottom left and middle left plots), the reference diode signal and PMT temperature fluctuations remain essentially within the range of 1-sigma, giving a relative variation of 0.2% and 1.8% in the lamp voltage and the PMT temperature, respectively. Translated onto the relative deviation of the measurements in both channels based on an average hydroperoxide signal of the liquid standard (5 ppbv), the average uncertainty arising from the

temperature-dependent noise is assumed to be maximally 1.6%. In order to account for potentially higher temperature-dependent signal drifts, frequent background measurements are performed during the measurement flights.

Since the dual-enzyme system is based on a wet-chemical measurement setup, we checked whether there is an impact of the dynamic flight patterns on the liquid transport during various maneuvers. In order to analyze the impact of aircraft movement in detail, a temporal series of the measured background signals in both channels was converted to hydroperoxide mixing ratios

under consideration of the aircraft's roll angle and body pitch rate (Fig. 4). Tracked latitude and longitude were plotted in order



to distinguish between actual flight maneuvers and stabilization processes of the aircraft due to turbulence, ascent, and descent maneuvers.

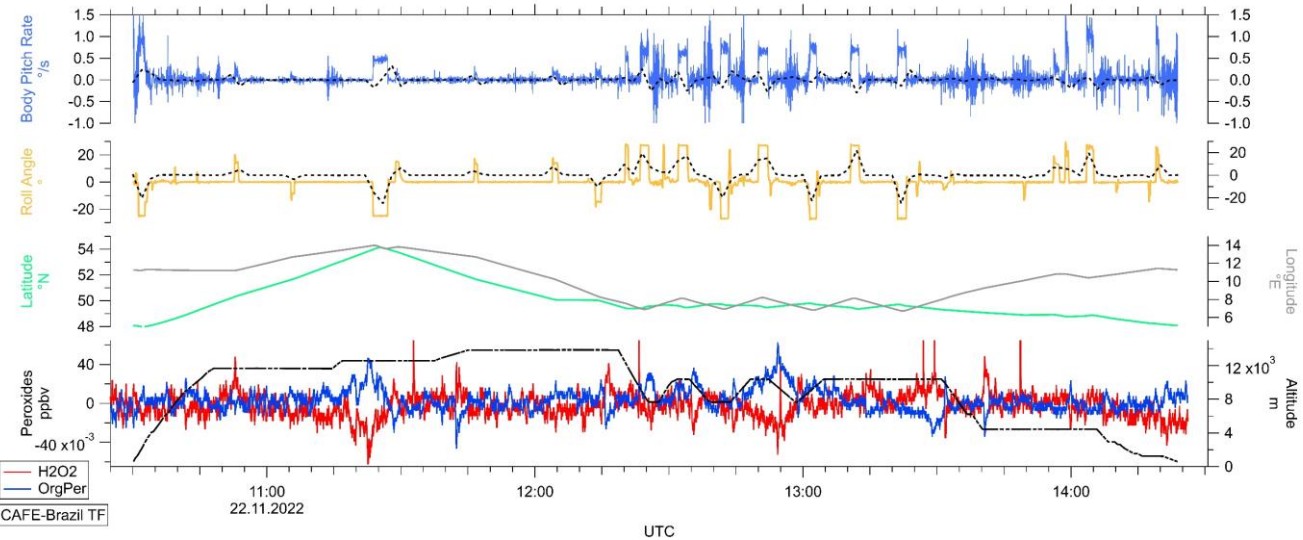

**Figure 4: Temporal series of the measured signals in channel A (H₂O₂ + ROOH; red) and B (ROOH; dark blue; bottom plot) relative**
**to the GPS flight altitude (black), latitude (green), longitude (grey), roll angle (yellow) and body pitch rate (blue; top plot) of the aircraft during the test flight of the CAFE-Brazil campaign performed on November 22, 2022 with 1-Hz measurement frequency. Dashed lines (black) represent the temporal trends of the roll angle and the body pitch rate based on 2-min bins.**

The signals of the measured background correlate with changes in the roll angle of the aircraft (yellow plot). Further, the signals harbor additional irregular noise of up to ± 20 pptv, which is most likely initiated by body pitch rate changes of the
aircraft during flight level alterations and turbulence. A comparable, but far more distinct periodic fluctuation of the background mixing ratios was observed during the test flights during the OMO-EU campaign in January 2015 (Fig. S3 in the Supplement). During frequent turns, the roll angle of the aircraft changes, which might result in motion of the measurement solutions in the liquid containers. In effect, air might be transported into the liquid system and accumulate in the flow-through cuvettes, where the motion might then be translated into characteristic periodic signal fluctuations due to variation in the
density-dependent extinction coefficient. Fig. S4 in the Supplement displays the flight pattern of the performed test flight with respect to the measured background signals and the pitch and roll angles of the aircraft. While the pitch angle modifications during descent and ascent maneuvers do not seem to correlate with the variations of the measured hydroperoxide mixing ratios, there is a clear connection between the extremes in the measured mixing ratios and the most significant changes in the aircraft roll angle. Consequently, the most impactful factors seem to be the roll angle changes and body pitch rate adjustments due to
turbulence, ascent, and descent of the aircraft. As shown in Fig. 4 and Fig. S3, the corresponding noise is assumed to be minor (up to 40 pptv for a few seconds during extended constant spiraling maneuvers) and does not seem to significantly affect the hydroperoxide data in the final temporal resolution of about 120 sec (Fig. S5).

Apart from the liquid transport within the instrument, also the ambient air transport might be significantly affected by the flight maneuvers. Fig. 5 gives an overview of the instrumental line and inlet pressures measured in the sampling line by the CPI



control unit and at the inlet manifold, respectively, the air mass flow tracked by the CPI internal unit, and the aircraft body pitch rate, which gives an overview on the performed descent and ascent maneuvers.

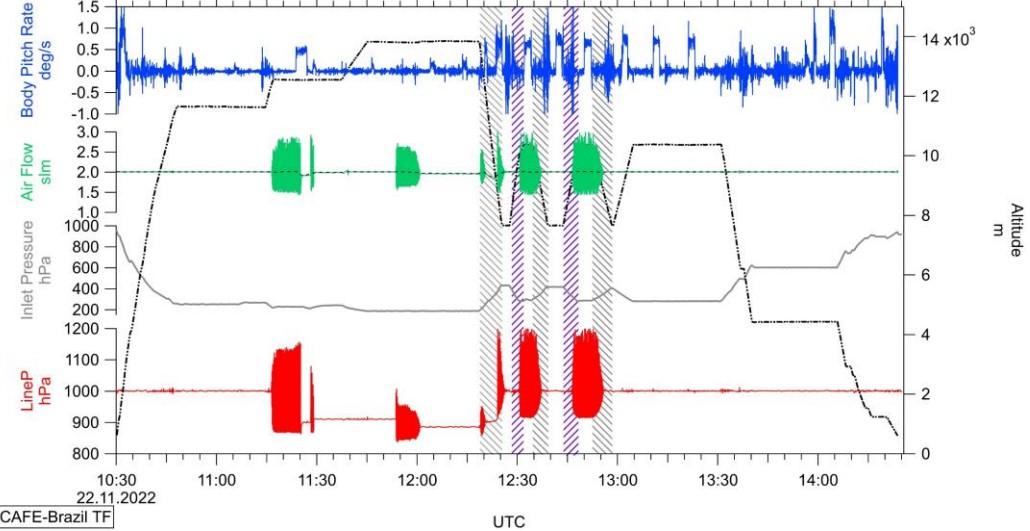

**Figure 5: Temporal series of the tracked line pressure (red) complimented by the GPS flight altitude (black), measured inlet pressure (grey), the air mass flow (green), and body pitch rate (blue) of the aircraft during the test flight of the CAFE-Brazil campaign performed on November 22, 2022 with 1-Hz measurement frequency. Rapid descent and ascent rates of the aircraft (2,000–3,000 ft/min) are highlighted in grey and purple, respectively.**

The sampling line pressure variations occur on one side during high-altitude flight legs as well as during rapid aircraft descent and ascent with rates of 2,000 ft/min or higher (maneuvers highlighted in grey and purple during the descent and ascent maneuvers, respectively). Additionally, atmospheric turbulence in close proximity to the jet stream reported during the lower flight legs (12.15–13.00 UTC) seem to affect the stability of the line pressure. Furthermore, from previous airborne measurements, we know that especially at high altitudes valve switches might induce line pressure fluctuations as well.

The CPI system seems to be triggered by the high airflow variability at the valves and adjustments of the pump speed, which results in an oscillating sampling line pressure. Most likely, during turbulence and rapid aircraft descent and ascent, the valve needles shift and alter the airflow in and/or out of the sampling line, leading to peaks in the detected line pressure and the described airflow regulation problems in the CPI system. However, the comparison of the background signals measured in both channels during the line pressure fluctuations does not correlate with short-term airflow fluctuations. Furthermore, no considerable measurement disturbances were observed during measurement flights where the line pressure fluctuations occurred, as exemplified in the Supplement of this work (Fig. S6). The high fluctuation frequency paired with the instrument's temporal resolution means that the average line pressure and air mass flow (975 ± 43.5 hPa and 2.00 ± 0.02 slm, respectively based on a 120 sec time sample) and, thus, the sample volume, do not seem to vary enough to affect the measured hydroperoxide mixing ratios significantly.



## 2.4 Instrument characterization: precision, limit of detection, temporal resolution, and total measurement uncertainty

According to the International Union of Pure and Applied Chemistry (IUPAC), the instrument detection limit (IDL) is defined as the smallest amount of an analyte, which produces a statistically significantly higher signal than the blank signal (Gold,

2019). Applied to our measurement method, the limit of detection is defined as the hydroperoxide mixing ratio, which can be distinguished from the background with a certainty of at least 95%. Analysis of the IDL is performed based on the 2-sigma uncertainty of the background measurements frequently performed during the measurement flights. The 2-sigma standard deviations ($2\sigma_{A,0}$, $2\sigma_{B,0}$) are calculated for both channels separately by determining the values for a background sample of approximately 200 points (at a resolution of 1 Hz). The limit of detection is consequently calculated by applying the subsequent

inlet transmission efficiencies ($ITE_{H_2O_2}$, $ITE_{ROOH}$) and the catalase destruction efficiency ($\varepsilon$) on the determined values:

$$IDL_{H_2O_2} = \sqrt{\left(\frac{2\sigma_{A,0}}{ITE_{H_2O_2}}\right)^2 + \left(\frac{2\sigma_{B,0}\cdot\varepsilon}{ITE_{ROOH}}\right)^2} \qquad (7)$$

A corresponding calculation is performed for the limit of detection of ROOH. Here, due to the lack of detailed information on organic hydroperoxide composition in the sampled ambient air, an additional correction factor based on the sampling efficiency of MHP (0.6) is incorporated:

$$IDL_{ROOH} = \frac{2\sigma_{B,0}}{0,6 \cdot ITE_{ROOH}} \qquad (8)$$

The detection limit of the instrument based on the $2\sigma$ uncertainty of 101 background measurements during the most recent measurement campaign, CAFE-Brazil, was determined to be 0.020 ppbv and 0.019 ppbv for $H_2O_2$ and ROOH, respectively. The instrument precision characterizes random errors and thus illustrates the proximity of the performed airborne measurements relative to each other. The instrument precision is determined from the reproducibility of 15 liquid calibrations

during CAFE-Brazil. Thus, it accounts for instrument noise, sensitivity drifts, and errors in the preparation of the liquid standards. Applied to hydroperoxide measurements, the precision is based on the 1-sigma uncertainty of the respective signals in the first phase of the calibration procedure. The hydrogen peroxide precision of the instrument is then determined using the resulting 1sigma standard deviations of both channels ($\sigma_{S,A}$, $\sigma_{S,B}$), corrected by the corresponding inlet transmission efficiencies ($ITE_{H_2O_2}$, $ITE_{ROOH}$) and the catalase destruction efficiency ($\varepsilon$). The organic hydroperoxide precision is based on

the signal standard deviation in channel B during the first calibration phase ($\sigma_{S,B}$) and the corresponding channel sensitivity ($ITE_{ROOH}$) and is extended by the sampling efficiency of 0.6.

$$P_{H_2O_2} = \sqrt{\left(\frac{\sigma_{S,A}}{ITE_{H_2O_2}}\right)^2 + \left(\frac{\sigma_{S,B}\cdot\varepsilon}{ITE_{ROOH}}\right)^2} \qquad (9)$$

$$P_{Org.HP} = \frac{\sigma_{S,B}}{0,6 \cdot ITE_{ROOH}} \qquad (10)$$

During the most recent measurement campaign over the Amazon region, i.e., CAFE-Brazil, the precision of the instrument

was determined to be 6.4% at 5.7 ppbv and 3.6% at 5.8 ppbv for $H_2O_2$ and ROOH, respectively.



Despite HYPHOP's ability to detect trace gas species with a 1-Hz frequency, due to the wet chemical measurement approach, the instrument's time resolution is heavily dependent on the liquid transport velocity during operation. The temporal resolution is defined as the time required to track a significant increase of the measured hydroperoxide mixing ratios. The determination of HYPHOP's maximal temporal resolution is based on analysis of the time needed to flush the flow through cuvettes with
fresh solutions defined as three times the volume of the cuvette at a given liquid flow velocity. The determination of the minimal temporal resolution of the instrument is based on the time tracked between 10% and 90% of the signal rise and fall in both channels based on liquid calibrations and background measurements. For the purposes of a detailed study on the instrument time resolution, the 10/90-method was further applied to liquid calibration measurements at varying $H_2O_2$ standard concentrations as well as peak mixing ratio events during flight legs in the upper troposphere associated with convective
transport. Tab. S1. in the Supplement lists the means ($\pm 1\sigma$) of the determined values. Based on the performed analysis, the time resolution based on the 10/90 method is determined to be 114 ($\pm$ 15.2) sec. The highest temporal resolution based on the flow-through cuvette flushing is determined to be 52.5 ($\pm 2.32$) sec. Considering the maximum and minimum values of the instrumental time resolution the spatial resolution at the average cruise speed of the aircraft (192 $\pm$ 46.5 m·s⁻¹) is approximately 11.5–23 km during the most recent field experiment (CAFE-Brazil).
The monitor's total measurement uncertainty (TMU) is defined based on Gauss's law for the propagation of uncertainties. The expression incorporates both the systematic and statistically driven deviations occurring during the measurement process and distinguishes between the measured species, i.e., hydrogen peroxide and the sum of organic hydroperoxides separately. The calculation of the hydrogen peroxide TMU is based on the standard deviations arising from the inlet transmission efficiency ($\sigma(ITE_{H_2O_2})$), ozone interference ($\sigma(IO_3)$), standard solution ($\sigma(LqStd)$), and the determined instrument precision ($P_{H_2O_2}$).
The corresponding total measurement uncertainty of ROOH is calculated based on the estimated instrument precision ($P_{ROOH}$) and the errors of the inlet and solution transmission efficiencies ($\sigma(ITE_{H_2O_2})$, $\sigma(STE_{ROOH})$).

$$TMU_{H_2O_2} = \sqrt{P_{H_2O_2}{}^2 + \sigma(LqStd)^2 + \sigma(ITE_{H_2O_2})^2 + \sigma(IO_3)^2} \tag{11}$$

$$TMU_{ROOH} = \sqrt{P_{ROOH}{}^2 + \sigma(STE_{ROOH})^2 + \sigma(ITE_{H_2O_2})^2} \tag{12}$$

The instrument precision (P) based on the 1σ standard deviation of the respective signals in the first phase of the calibration
procedure contributes further to the uncertainty of the standard (LqStd). The uncertainty of the inlet efficiency (ITE) is based on the 1σ standard deviation from the inlet sampling efficiencies for hydrogen peroxide and the sum of organic peroxides. The uncertainty of the ozone interference ($IO_3$) is derived from the 1σ standard deviation of the linear regression of $H_2O_2$ versus $O_3$. The inlet efficiency uncertainty was determined to be 1.6% for $H_2O_2$ and 1.1% for ROOH. The uncertainty of $O_3$ interference was determined under laboratory conditions as 10.2%. During the CAFE-Brazil campaign, the total measurement
uncertainties of $H_2O_2$ and ROOH were determined to be 12% and 40%, respectively. The determined parameters are comparable with those of previous campaigns, i.e., OMO (Hottmann et al., 2020), CAFE-Africa (Hamryszczak et al., 2023), and BLUESKY (Hamryszczak et al., 2022), as presented in Tab. S2. Considering the entirety of performed airborne



measurements between 2015 and 2023, the TMU varied between 9–28% and 40–41%, with precision of up to 1.3% (at 5.9 ppbv) and 6.4% (at 5.7 ppbv) for $H_2O_2$ and ROOH, respectively. The instrument limit of detection varied within the range of

8–53 pptv for $H_2O_2$ and 6–52 pptv for ROOH, respectively. Further, based on the all the datasets from the previous measurement campaigns, the temporal resolution was determined to be 120 sec. Further potential optimization of the temporal resolution can be performed by reducing the dead volume of the liquid supply through reduction of the tubing length or by optimization of the peristaltic pump speed at the potential cost of an increased LOD.

## 3 Performance of HYPHOP during CAFE-Brazil

During the CAFE-Brazil aircraft campaign in December 2022 and January 2023, 20 measurement flights were conducted, mostly over the Amazon Basin in Brazil (12 °S–4 °N; 70–38 °W). The main objective of the field experiment was to investigate the distribution of trace gases, aerosols, and radicals under pristine conditions in contrast to environments affected by anthropogenic emissions. The campaign also focused on convective transport and the interactions between the tropospheric layers under different meteorological conditions, the interaction between the biosphere and the atmosphere from the

perspective of atmospheric chemical exchange processes, and cloud formation over the tropical rainforest. The flights were performed from the base of operation in Manaus (3° 6′ S, 60° 1′ W) and covered the altitudinal range between a few tens of meters above the surface and an altitude of approximately 15 km. Fig. S7 gives an overview of the performed flights during the campaign and the regions covered. The observed mixing ratios of hydroperoxides during CAFE-Brazil are presented in a latitude vs. longitude plot with mean mixing ratio values binned into a subset of 1° x 1° bins along the flight tracks for the

three tropospheric layers (0 < 2 km: top panel; 2 < 8 km: middle panel; ≥ 8 km: bottom panel; Fig. 4). The color scale represents the measured mixing ratios of $H_2O_2$ (left panel) and ROOH (OrgPer; right panel). Please note that for resolution purposes the color scaling varies between the panels.


**Figure 6: Spatial distribution of measured hydrogen peroxide (left panel) and the sum of organic hydroperoxides (right panel) in the boundary layer (top panel), middle troposphere (middle panel), and the upper troposphere (≥ 8 km; bottom panel) during the CAFE-Brazil campaign. Data were binned into a 1° x 1° subset over the flight tracks based on a 1-Hz measurement frequency. Please note that for resolution purposes the color scaling differs between plots. Global coastline and global country boundaries are based**

**on data set available from WaveMetrics. [1]**

---

[1] WaveMetrics, Inc. 10200 SW Nimbus, G-7 Portland , OR 97223;





The mean ($\pm 1\sigma$) and median mixing ratios based on all measured $H_2O_2$ and ROOH species during the campaign were 0.30 ($\pm 0.30$) ppbv, 0.17 ppbv and 0.43 ($\pm 0.36$) ppbv, 0.28 ppbv, respectively, with maximum mixing ratios reaching 1.94 ppbv and 1.73 ppbv for hydrogen peroxide and the total organic hydroperoxides, respectively. Please note that the mean and median hydroperoxide mixing ratios were calculated based on measurement results with the instrumental measurement

frequency of 1 Hz. Table S3 in the Supplement gives an overview of $H_2O_2$ and ROOH mean ($\pm 1\sigma$) and median mixing ratios with the corresponding maximum peroxide levels for each tropospheric layer.

The highest mean ($\pm 1\sigma$) and median mixing ratios were detected between altitudes of 0 and 2 km. The mean and median hydroperoxide levels decrease progressively with increasing altitude, with the lowest mixing ratios measured in the upper troposphere (UT; above 8 km). Especially above 2 km, a trend of increasing hydroperoxide levels from the coastal area towards

the central Amazon Forest was observed. This may be expected due to the influence of vegetation-derived VOC emissions. Generally, ROOH levels appear to be approximately a factor of 1.5 higher relative to $H_2O_2$ throughout the entire troposphere. The high ROOH mixing ratios are most likely due to an efficient production based on vegetation emissions and the sufficient availability of $HO_2$ radicals derived from the photolysis of carbonyls and the photooxidation of VOCs and CO by OH radicals. Additionally, hydrogen peroxide is expected to be affected to a larger extent by deposition processes due to its higher solubility

and uptake by vegetation.

Slightly higher hydroperoxide levels were observed in the upper troposphere as a result of air masses originating from the South Atlantic Convergence Zone (SACZ), where average mixing ratios of 0.15–0.25 ppbv and 0.25–0.40 ppbv for $H_2O_2$ and ROOH, respectively, over a 1° x 1° bin of merged data were observed. These maxima are most likely due to the atmospheric transport of hydroperoxides or precursors to the upper troposphere. Locally, peaking hydroperoxide levels in the UT were

further observed in the northeastern and southwestern parts of the sampled region. Here, the mean mixing ratios were between 0.35 ppbv and 0.55 ppbv with maxima of up to 0.40 ppbv and 0.56 ppbv for $H_2O_2$ and ROOH, respectively. Based on the flight log book, strong convective activity during the respective measurement flights was reported in the sampled regions. The overall mean vertical hydroperoxide distribution over the sampled region is displayed in Fig. 7.

---

Data access under: https://www.wavemetrics.net/Downloads/IgorGIS/GISData/; <last access: 09.06.2023>

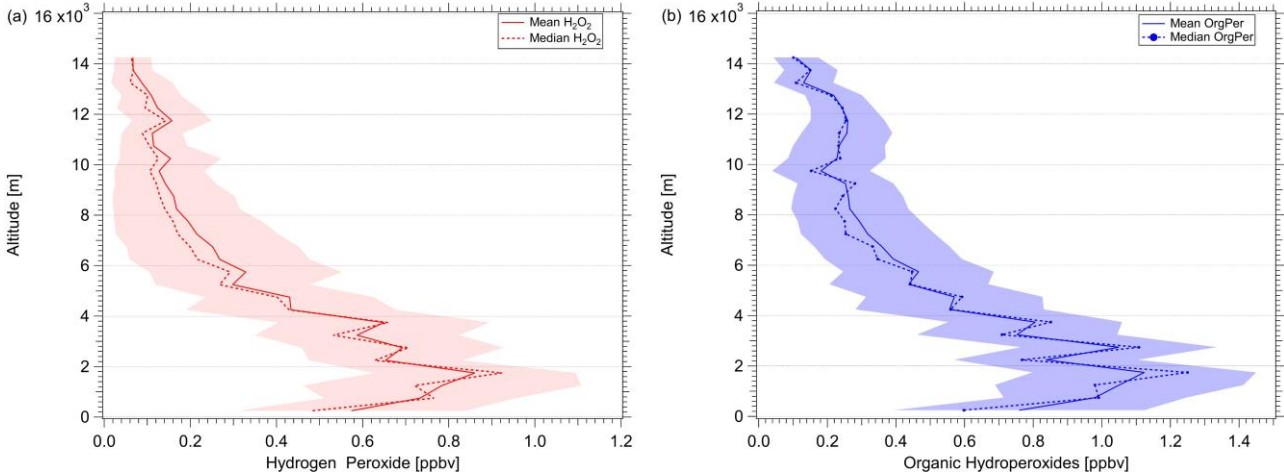

**Figure 7: Vertical profiles of H₂O₂ (red; a) and ROOH (blue; b). Vertical profiles were calculated as means and medians over 500-m layers over the atmospheric column based on data with 1-Hz measurement frequency obtained in the sampled region.**

In general, the observed vertical hydroperoxide distributions follow the expected trends throughout the troposphere (Fig. 7). The lowest hydroperoxide mixing ratios ($\pm 1\sigma$) of 0.07 ($\pm 0.04$) ppbv for $H_2O_2$ and 0.11 ($\pm 0.06$) ppbv for ROOH were measured in the upper troposphere. The highest mean values ($\pm 1\sigma$) of 0.86 ($\pm 0.23$) ppbv and 1.12 ($\pm 0.33$) ppbv for $H_2O_2$ and ROOH,

respectively, were measured directly above the boundary layer (~ 2 km) in the lower free troposphere. Below 1 km, hydroperoxide levels decrease to 0.57 ($\pm 0.26$) ppbv and 0.75 ($\pm 0.36$) ppbv for $H_2O_2$ and the organic hydroperoxides, respectively, reflecting the impact of deposition processes involving vegetation in the boundary layer.

The hydroperoxide vertical profiles show increased levels in the UT (10–13 km), which might indicate convective outflow. Here, the mixing ratios increase to approximately 0.16 ($\pm 0.09$) ppbv and 0.26 ($\pm 0.11$) ppbv for $H_2O_2$ and ROOH, which is

approximately 19% and 23%, respectively, corresponding to the measured maximum hydroperoxide mixing ratios directly above the boundary layer. The comparison of $H_2O_2$ and ROOH vertical profiles shows a high abundance of ROOH throughout the entire tropospheric column relative to $H_2O_2$. Especially in the UT, the ROOH mixing ratios seem to increase significantly in comparison to $H_2O_2$ (up to a factor of 5). $H_2O_2$ is temporarily or permanently removed by wet deposition in the lower part of convective clouds; the influence of wet deposition on the transport of the less soluble ROOH is expected to be smaller.

Additional processes responsible for the high levels of measured organic peroxides might be of significant importance to the local budget of ROOH, which in turn can contribute to the local $HO_x$ budget. Over the pristine Amazon forest, a large suite of VOCs, including highly oxygenated organic molecules (HOMs), which are an additional source of organic hydroperoxides in the UT, might be transported via convection from the lower troposphere. Further analysis of these data, incorporating results with additional trace gases, will be the subject of future work.





# 4. Conclusions

In this work, we introduced a hydroperoxide measurement setup designed for operation on board of the research aircraft HALO. The presented wet chemical, dual-enzyme measurement method enables us to perform *in situ* measurements up to the tropopause and lower stratosphere (1000–150 hPa). Potential chemical interference affecting the airborne hydroperoxide measurements can be easily eliminated or corrected by simply adding reactants (EDTA; HCHO) or in the data acquisition process. Based on a test flight specifically dedicated to investigating measurement inconsistencies arising from the dynamic flight patterns of the aircraft, we analyzed potential noise sources affecting the background signal. Despite technical challenges during the dynamic flight maneuvers, the majority of these issues do not have a significant impact on the determination of the absolute hydroperoxide mixing ratios during background measurements. The instrument shows some periodic fluctuations, which are most likely induced by the roll angle changes during the flight maneuvers. The periodically increased noise during the performed experiment flight was estimated to be in the approximate range of the IDL. The observed periodic variation of the signal is assumed to only have a minor impact on the hydroperoxide measurements due to significantly higher levels of this species in the troposphere. Changes in cabin temperature exerted an impact on noise at the lowest mixing ratios, which is likewise not considered to affect the measurements. Rapid line pressure variations due to temporal pressure instabilities, most likely caused by valve needle operation inconsistencies, were also observed during the high flight legs as well as during rapid descent and ascent maneuvers. No significant impact on the signals in either of the channels and thus on the measurements was observed. Due to the determined instrument temporal resolution of approximately 120 sec, the high-frequency airflow fluctuations are assumed not to affect the sampling volume significantly and thus likely have no impact on the measured mixing ratios of hydroperoxides. However, instrument performance can be further optimized by minimizing the technical disturbances during the measurement flights. Potential higher temperature-dependent signal drifts are corrected by frequent background examination on the consecutive flight legs during the measurements. Furthermore, the instrument time resolution can be increased by reducing the length of the liquid supply tubing. Analysis of potential noise sources impacting the *in situ* measurements shows that the hydroperoxide data is of sufficient quality for atmospheric studies. Based on the most recent airborne measurements during the CAFE-Brazil aircraft campaign, HYPHOP faithfully captures the temporal and spatial variability of hydroperoxides in the troposphere. Dynamic processes such as convection, cloud scavenging, and subsequent rainout or horizontal transport of the hydroperoxides are sufficiently well captured by the instrument through the entire tropospheric column and along the long-range measurement flight tracks.

**Data availability.** All data measured during the CAFE-Brazil campaign are available upon request.

**Author contributions.** HF and ZH designed the study; JL and HF planned the campaign; ZH, DD, and AH performed the measurements; ZH, DD, and AH processed and analyzed the data; RK, SH, and HF designed the instrument; RK, UP, SH, BB, DD, and ZH optimized the instrument; ZH wrote the manuscript draft with contributions of all co-authors.



**Competing interests.** The authors declare that they have no conflicts of interest.


**Acknowledgments.** The authors are very grateful to the CAFE-Brazil team, Forschungszentrum Jülich, Karlsruhe Institute of Technology and Deutsches Zentrum für Luft- und Raumfahrt (DLR) in Oberpfaffenhofen, whose support was essential for the project.

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
