# Peer review of "HYPHOP: a tool for high-altitude, long-range monitoring of hydrogen peroxide and higher organic peroxides in the atmosphere"

_Atmospheric Measurement Techniques, 2023_

## Referee Comment (RC2)

Review of Hamryszczak et al., AMTD (2023):

The manuscript presents instrumental details for HYPHOP, a dual-enzyme fluorescence spectroscopy-based measurement of hydrogen peroxide and organic peroxides from aircraft. Chemical and environmental challenges to the measurement are discussed and quantified in the context of a 2022-2023 field campaign in Brazil, and data from that campaign are presented in very general terms. Considering that the instrument has flown on multiple field campaigns since 2015, describing it in the literature is valuable exercise. The manuscript needs some revisions before it is ready for publication.

One major concern relates to instrument precision, accuracy, and how best to convey those values. I have a few related comments:

1. When you report data for a field project, do you share 1 Hz data or 120 s data? I found the intermixing of the two time bases to be confusing. Instrument precision, etc., should be presented for the time basis of the mixing ratio data that you provide for field projects.
2. Instrument precision (in the abstract and section 2.4) is presented for values near 6 ppbv. These values are much too high to convey instrument precision relevant to your field measurements when median mixing ratios observed during CAFE-Brazil were 170 pptv and 280 pptv for $H_2O_2$ and organic peroxides, respectively, and maximum values were 1.94 ppbv and 1.73 ppbv.
3. The "precision" mentioned in comment 2 seems more appropriately described as a component of measurement accuracy because it largely consists of the reproducibility of laboratory calibration experiments. Consider reevaluating how you describe these attributes.

Throughout the manuscript text and Figure 4, I suggest using pptv instead of ppbv for values such as 0.020 ppbv to improve readability.

How do the measurement performance characteristics (e.g., limit of detection, precision, accuracy, time resolution) of your instrument compare with the other aircraft instruments you mention in the introduction? A bit of comparison would help provide context for your reader.

In the introduction where you first mention details such as "…a compact V25 control unit…" and "…a constant pressure inlet unit (CPI)…", the reader really has no context to make this information useful. Consider adding text that refers the reader to the appropriate section later in the paper where these terms are better described, e.g., "…as described in Section 2.1…"

Lines 101-102: What is a "manually cooled liquid container department"?

Line 106: You mention constant pressure here but don't mention a value until line 125 "…approximately ambient pressure…"— consider giving a value for the pressure control at line 106.

Line 112: With a forward-facing inlet, how do particles and cloud droplets impact your measurement? Do you remove data affected by condensed phase material?

Line 117: What airflow does the pump provide at maximum altitude (150 hPa ambient)?

Line 126: I assume that the "reaction coil" mentioned here is the "stripping coil" shown in Figure 2. Making the body text match the figure text would be helpful.

Line 127 and elsewhere: You mention precooled solutions but nowhere in the manuscript do you give details on how the temperature control of the cooling is performed and how the temperatures of the solutions are monitored. Considering how important temperature is to your assumed sampling efficiencies, these details seem very important to the performance of your instrument and are critical to include in this manuscript.

In Figures 3, 4, 5, and 7: consider changing the altitude unit from m to km to get rid of the "$x10^3$."

Lines 159-161: Calibrating with a liquid $H_2O_2$ standard calibrates the measurement to a certain extent, but it misses important stages of your instrument. Have you ever calibrated with gas phase peroxides? At multiple mixing ratios (or at least multiple $H_2O_2$ solution concentrations) to verify linearity?

Lines 162-163: Why is the HCl wash needed to precede the calibrations but not the ambient measurements?

Lines 170-190 and related parts of the Supplement: All variables need to be defined and units given for each.

Lines 180-184: What are typical values for the destruction efficiency?

Line 190 through the end of Section 2.2: It would be good to introduce this section by naming the individual corrections and giving typical values (in pptv) for them.

Lines 191-193: How much does the background typically vary, in terms of pptv of peroxide, between background measurements? Over the course of a flight?

Lines 199-200: Do you expect the inlet transmission efficiency to be temperature and pressure invariant?

Line 200: Regarding flow rates, 30 slm and 10 slm at all altitudes or is there a range of flow rates?

Line 205: Using MHP instead of PAA for your inlet and CPI pump transmission tests would seem preferred, since MHP is your assumed organic peroxide.

Line 221: Is the temperature of the sampling solution measured just prior to the reaction coil? Have you measured these sampling efficiencies with your instrument? And what would the sampling efficiencies be for PAA or HMHP?

Lines 228-229: The transmission efficiency values would fit better at the end of the previous paragraph. Are these for the inlet + CPI pump or without the pump? It would be valuable to state both sets of values.

Lines 239-241: have you experimentally confirmed that these additions do indeed eliminate all of the interferences?

Lines 253-254: Delete "…but also cabin…" and replace with "and"

Lines 255-263: This test is valuable but really only evaluates changes in background, not changes in instrument sensitivity, since no calibration standard was added. Are the flight maneuver effects expected to only affect background levels, not instrument sensitivity?

Line 275-280: 5 ppbv does not seem to be the appropriate reference here—shouldn't it be the signal levels of typical ambient mixing ratios since that's where the temperature variations are relevant (not during calibrations)? Using 5 ppbv badly underestimates the uncertainty caused by temperature-dependent noise, unless I'm missing something.

Lines 323-325: Are the valves mounted vertically or horizontally? Some researchers have found improved immunity to aircraft motion by mounting valves horizontally.

Line 335: Change "…mixing ratio, which can be…" to "…mixing ratio that can be…"

Lines 249-352: As mentioned earlier, reproducibility of calibrations is more appropriately considered as part of instrument accuracy rather than precision. Precision itself will presumably be a function of mixing ratio and would be best quoted at a value much closer to ambient mixing ratios than the calibration standard value. Regarding accuracy and total measurement uncertainty, how well do you know the absolute value of your calibration standard?

Line 468: Considering the significant ozone interference and assumption of zero $H_2O_2$ in the LS to determine the interference value, a qualifier might be appropriate for the claim of measurements into the lower stratosphere.

---

## Author Comment (AC1)

**Answer to: Comment on "HYPHOP: a tool for high-altitude, long-range monitoring of hydrogen peroxide and higher organic peroxides in the atmosphere" by Hamryszczak et al.**
**Anonymous referee #1**

**Please note the used color code**
**(black: Referee Comments, red: Author Comments, blue: manuscript changes according to Referee's recommendations and comments)**

Review comments for: HYPHOP: a tool for high-altitude, long-range monitoring of hydrogen peroxide and higher organic peroxides in the atmosphere

Hamryszczak et al. detail the performance of the HYPHOP instrument for airborne hydroperoxide measurements. The instrument has already been used in a number of field experiments and a full description in the literature is warranted. This manuscript is a thorough description of the instrument and its performance. Multiple sources of measurement uncertainty are characterized including chemical interference, dynamic flight patterns, cabin temperature, and line pressure changes. I recommend publication after attention to the following minor comments.

We thank the referee for his/her valuable comments and recommendations. Following the referee's recommendation, the manuscript was changed as described in detail below.

Comments:

The introduction/manuscript is lacking information about the performance of other airborne hydroperoxide measurement techniques. How does HYPHOP performance compare to the best available alternatives?

The following table gives a general overview regarding the measurement performance of the most common hydroperoxide measurement techniques with their corresponding references. The table was added to the supplementary information of the manuscript (Table S1 of the Supplement). Corresponding brief reference was added in L56.

**Table S1: Performance comparison of the most common hydroperoxide measurement techniques relative to the HYPHOP monitor (respective performance parameters are based on Kleindienst et al. 1988; Mackay et al. 1990; Staffelbach et al. 1996; Crounse et al. 2006; St Clair et al. 2010; Allen et al. 2022).**

| | HYPHOP | HPLC | TDLAS | CIT-CIMS |
|---|---|---|---|---|
| **Sampling interval** | continuous | 45 min | 60 sec | continuous |
| **Data point frequency** | 1 Hz | $0.28 \cdot 10^{-3}$ Hz | $0.56 \cdot 10^{-3}$ Hz | > 1 Hz |
| **Instrumental detection limit (IDL)** | $H_2O_2$: 20 pptv
ROOH: 19 pptv | $H_2O_2$: 150 ppt$_v$
ROOH: 30 ppt$_v$ | $H_2O_2$: 100 ppt$_v$ | $H_2O_2$: 1–10 pptv
MHP: 1–10 pptv |
| **Precision** | $H_2O_2$: 360 pptv
ROOH: 210 pptv | $H_2O_2$: -
ROOH: - | $H_2O_2$: - | $H_2O_2$: 50 ppt$_v$
MHP: 50 ppt$_v$ |
| **Accuracy** | $H_2O_2$: 0.7%
ROOH: 0.8% | $H_2O_2$: -
ROOH: - | $H_2O_2$: 20% | $H_2O_2$: -
MHP: 40% |
| **Total measurement uncertainty (TMU)** | $H_2O_2$: 12%
ROOH: 40% | $H_2O_2$: 20%
ROOH: 20% | $H_2O_2$: 20% | $H_2O_2$: 35%
MHP: 40–80% |
| **Artifacts** | $O_3$
$SO_2$
Metal ions
(NO) | Pollution
Particles | none | $H_2O$
$HOCH_2OH$ |

L56 was changed to:
An overview of the measurement performance of the airborne measurement techniques discussed above relative to the instrument presented in the scope of this work is shown in the Supplement of this work (Tab. S1).

The HYPHOP background measurements are important for the calibration, but there is no mention of how often these measurements are taken during a typical flight (other than "frequent measurements") or how the background is interpolated.

We thank the referee for pointing out the missing information. L199–209 (former L190–193) was changed according to the referee's comment.

L199–209 (former L190–193) was changed to:
Due to the characteristics of the wet chemical measurement method, corrections regarding background signal variations (15–33 pptv between two consecutive background measurements and approximately 50–70 pptv over the duration of a typical measurement flight) and time modification due to the delay caused by liquid transport within the instrument (approximately 300 sec) have to be performed. Additionally, signal corrections regarding hydroperoxide transmission efficiencies due to potential wall losses at the inner surface of the sampling inlet (up to 300 pptv at 1.5 ppbv $H_2O_2$ and up to 100 pptv at 1.5 ppbv PAA, respectively), and sampling efficiencies have to be initially performed to obtain absolute hydroperoxide mixing ratios.
In order to account for potential measurement divergencies and background signal alterations initiated by pressure, and temperature instabilities during the measurement flights, the frequently measured background (typically at least 3–4 times per flight) signal is interpolated according to the background measurement signals ($U_{A,0}$; $U_{B,0}$) obtained during the four-point calibration procedure. A typical background sampling frequency and duration are presented based on an exemplary measurement flight during the most recent research campaign, CAFE-Brazil in the Supplement of this work (Fig. S3).

The performance of this instrument is evaluated in pristine air – a statement on any projected interfering variables in unclean air (ex. urban air, wildfires) would be beneficial.

The following table gives an overview of potential chemical interferences related to polluted air masses and biomass burning[1]. The overview was incorporated into the Supplement of the manuscript (Tab. S2). L85 (former L83) was extended by information referring to the presented table.

**Table S2: Overview of potential chemical interferences affecting the measurement performance of the HYPHOP monitor. The overview is based on the information provided by the commercial distributor of the instrument, on which the HYPHOP set up is based (Aero-Laser, Garmisch-Partenkirchen, Germany)[1].**

| Tropospheric trace gas | Max. expected interference |
|---|---|
| $O_3$ | 30pptv $H_2O_2$/100 ppbv |
| NO | 12 pptv $H_2O_2$/100 ppbv |
| PAN | X |
| $NO_2$ | X |
| Glyoxal | X |
| Isobutane | X |
| Isobutylene | X |
| 1-Butane | X |
| HCHO | X |
| Benzene | X |
| Toluene | X |
| MeOH | X |

[1] https://www.aero-laser.de/gas-analyzers/h2o2-al2021.html (last access: 27.07.23)

| | |
|---|---|
| **Acetone** | **X** |
| **Methylamine** | **X** |
| **Dimethylamine** | **X** |
| **n-Butane** | **X** |
| **Cis-2-Butene** | **X** |
| **Trans-2-Butene** | **X** |
| **Iodide** | **X** |
| **Chloride** | **X** |
| **Nitrate** | **X** |
| **Bromide** | **X** |
| **Phosphate** | **X** |
| **Benzoate** | **X** |

L85 (former L83) was changed to:

Further chemically driven interferences potentially affecting the instrumental measurement performance are not considered based on the information on the commercially available hydroperoxide monitor AL2021 (Aero-Laser, Garmisch-Partenkirchen, Germany), on which the HYPHOP monitor is based (Tab. S2).

**References**

Allen, Hannah M.; Crounse, John D.; Kim, Michelle J.; Teng, Alexander P.; Ray, Eric A.; McKain, Kathryn et al. (2022): H 2 O 2 and CH 3 OOH (MHP) in the Remote Atmosphere: 1. Global Distribution and Regional Influences. In *JGR Atmospheres* 127 (6). DOI: 10.1029/2021JD035701.

Crounse, John D.; McKinney, Karena A.; Kwan, Alan J.; Wennberg, Paul O. (2006): Measurement of gas-phase hydroperoxides by chemical ionization mass spectrometry. In *Analytical chemistry* 78 (19), pp. 6726–6732. DOI: 10.1021/ac0604235.

Kleindienst, T. E.; Shepson, P. B.; Hodges, D. N.; Nero, C. M.; Arnts, R. R.; Dasgupta, P. K. et al. (1988): Comparison of techniques for measurement of ambient levels of hydrogen peroxide. In *Environmental science & technology* 22 (1), pp. 53–61. DOI: 10.1021/es00166a005.

Mackay, G. I.; Mayne, L. K.; Schiff, H. I. (1990): Measurements of H 2 O 2 and HCHO by Tunable Diode Laser Absorption Spectroscopy During the 1986 Carbonaceous Species Methods Comparison Study in Glendora, California. In *Aerosol Science and Technology* 12 (1), pp. 56–63. DOI: 10.1080/02786829008959325.

St Clair, Jason M.; McCabe, David C.; Crounse, John D.; Steiner, Urs; Wennberg, Paul O. (2010): Chemical ionization tandem mass spectrometer for the in situ measurement of methyl hydrogen peroxide. In *The Review of scientific instruments* 81 (9), p. 94102. DOI: 10.1063/1.3480552.

Staffelbach, Thomas A.; Kok, Gregory L.; Heikes, Brian G.; McCully, Brian; Mackay, Gervase I.; Karecki, David R.; Schiff, Harold I. (1996): Comparison of hydroperoxide measurements made during the Mauna Loa Observatory Photochemistry Experiment 2. In *J. Geophys. Res.* 101 (D9), pp. 14729–14739. DOI: 10.1029/95JD02197.

---

## Author Comment (AC2)

**Answer to: Comment on "HYPHOP: a tool for high-altitude, long-range monitoring of hydrogen peroxide and higher organic peroxides in the atmosphere" by Hamryszczak et al.**
**Anonymous referee #2**

**Please note the used color code**
**(black: Referee Comments, red: Author Comments, blue: manuscript changes according to Referee's recommendations and comments)**

Review of Hamryszczak et al., AMTD (2023):

The manuscript presents instrumental details for HYPHOP, a dual-enzyme fluorescence spectroscopy-based measurement of hydrogen peroxide and organic peroxides from aircraft.
Chemical and environmental challenges to the measurement are discussed and quantified in the context of a 2022-2023 field campaign in Brazil, and data from that campaign are presented in very general terms. Considering that the instrument has flown on multiple field campaigns since 2015, describing it in the literature is valuable exercise. The manuscript needs some revisions before it is ready for publication.

We thank the referee for her/his very valuable comments and detailed recommendations. Following the referee's recommendation, the manuscript was changed as described in detail below.

One major concern relates to instrument precision, accuracy, and how best to convey those values. I have a few related comments:

1. When you report data for a field project, do you share 1 Hz data or 120 s data? I found the intermixing of the two time bases to be confusing. Instrument precision, etc., should be presented for the time basis of the mixing ratio data that you provide for field projects.

We apologize for the confusion. Both, calculations and the data reports are based on 1-Hz time basis. As elaborated in L361 ff. (now L376 ff.) due to restrictions caused by liquid transport of the instrument, a temporal instrumental resolution of approximately 120 sec is determined. L22–25 was changed according to the referee's comment.

L22–25 was changed to:
The instrument precision based on the measurement results during CAFE-Brazil for hydrogen peroxide and the sum of organic hydroperoxides is estimated to be 6.4% (at 5.7 ppbv) and 3.6% (at 5.8 ppbv), respectively, and the corresponding detection limits 20 pptv and 19 pptv for a data acquisition frequency of 1 Hz. The determined instrumental temporal resolution is given at approximately 120 sec.

2. Instrument precision (in the abstract and section 2.4) is presented for values near 6 ppbv. These values are much too high to convey instrument precision relevant to your field measurements when median mixing ratios observed during CAFE-Brazil were 170 pptv and 280 pptv for $H_2O_2$ and organic peroxides, respectively, and maximum values were 1.94 ppbv and 1.73 ppbv.

We agree with the referee regarding the initial difficulties arising from the discrepancy between the mixing ratios used to determine the instrumental precision and the actually measured ambient mixing ratios during the CAFE-Brazil campaign. However, the determined instrumental precision based on the reproducibility of the liquid standard mixing ratios serves here as the upper limit, whilst the lower limit of the instrumental precision is given by the instrumental detection limit (IDL). Additionally, for the purposes of comparability with the measurement results during former airborne campaigns i. e. OMO, CAFE-Africa and BLUESKY, during which substantially higher hydroperoxide mixing ratios were locally detected relative to the results of CAFE-Brazil ROOH mixing ratios using the HYPHOP monitor, the presented approach is convenient.

3. The "precision" mentioned in comment 2 seems more appropriately described as a component of measurement accuracy because it largely consists of the reproducibility of laboratory calibration experiments. Consider reevaluating how you describe these attributes.

We apologize for the confusion. The chosen approach to determine instrumental precision is based on the information regarding the reproducibility of the mixing ratios of the liquid standard performed prior to each measurement flight under campaign conditions (by no means performed in a laboratory environment) over the entire measurement period of the campaign. The accuracies with respect to the absolute hydroperoxide concentrations (determined via titration of the stock solution with potassium permanganate) based on the performed calibrations are determined to be on average 0.7% (at 5.7 ppbv) for $H_2O_2$ and 0.8% (at 5.7 ppbv) for ROOH, respectively.

Throughout the manuscript text and Figure 4, I suggest using pptv instead of ppbv for values such as 0.020 ppbv to improve readability.

The manuscript text was changed according to the referee's recommendation. Please note that for purposes of convenience, we postpone providing a detailed list of changed lines in this comment. Instead, the announced changes can be tracked to full satisfaction in the uploaded Author's tracked changes version of the manuscript.

How do the measurement performance characteristics (e.g., limit of detection, precision, accuracy, time resolution) of your instrument compare with the other aircraft instruments you mention in the introduction? A bit of comparison would help provide context for your reader.

The following table presents an overview of the most common hydroperoxide measurement techniques and their corresponding references. The table was added to the supplementary information of the manuscript upon the referee's recommendation (Table S1 of the Supplement). A corresponding reference was added in L56.

**Table S1: Performance comparison of the most common hydroperoxide measurement techniques relative to the HYPHOP monitor (respective performance parameters are based on Kleindienst et al., 1988; Mackay et al., 1990; Staffelbach et al., 1996; Crounse et al., 2006; St Clair et al., 2010; Allen et al., 2022).**

|  | HYPHOP | HPLC | TDLAS | CIT-CIMS |
|---|---|---|---|---|
| **Sampling interval** | continuous | 45 min | 60 sec | continuous |
| **Data point frequency** | 1 Hz | $0.28 \cdot 10^{-3}$ Hz | $0.56 \cdot 10^{-3}$ Hz | > 1 Hz |
| **Instrumental detection limit (IDL)** | $H_2O_2$: 20 pptv ROOH: 19 pptv | $H_2O_2$: 150 $ppt_v$ ROOH: 30 $ppt_v$ | $H_2O_2$: 100 $ppt_v$ | $H_2O_2$: 1–10 pptv MHP: 1–10 pptv |
| **Precision** | $H_2O_2$: 360 pptv ROOH: 210 pptv | $H_2O_2$: - ROOH: - | $H_2O_2$: - | $H_2O_2$: 50 $ppt_v$ MHP: 50 $ppt_v$ |
| **Accuracy** | $H_2O_2$: 0.7% ROOH: 0.8% | $H_2O_2$: - ROOH: - | $H_2O_2$: 20% | $H_2O_2$: - MHP: 40% |
| **Total measurement uncertainty (TMU)** | $H_2O_2$: 12% ROOH: 40% | $H_2O_2$: 20% ROOH: 20% | $H_2O_2$: 20% | $H_2O_2$: 35% MHP: 40–80% |
| **Artifacts** | $O_3$ $SO_2$ Metal ions (NO) | Pollution Particles | none | $H_2O$ $HOCH_2OH$ |

L56 was changed to:
An overview of the measurement performance of the airborne measurement techniques discussed above relative to the instrument presented in the scope of this work is shown in the Supplement of this work (Tab. S1).

In the introduction where you first mention details such as "...a compact V25 control unit..." and "...a constant pressure inlet unit (CPI)...", the reader really has no context to make this information useful. Consider adding text that refers the reader to the appropriate section later in the paper where these terms are better described, e.g., "...as described in Section 2.1..."

L67–71 (former L65–68) were changed based on the referee's recommendation.

L67–71 (former L65–68) were changed to:
HYPHOP was designed as a lightweight and compact monitor, where the measurement process and the complementary data acquisition are controlled by a compact V25 control unit (Max Planck Institute for Chemistry, Mainz, Germany), described in more detail in Sect. 2.2. As will be discussed in Sect. 2.1., the instrument was further equipped with a constant pressure inlet unit (CPI), making it suitable for *in situ* measurements under varying ambient air pressures.

Lines 101-102: What is a "manually cooled liquid container department"?

The manually cooled container department is a tidily shielded rack section dedicated to storage of the analytical liquids cooled by Thermosafe (U-tec) cooling packs at temperatures of -1 – -23°C maintaining liquid temperatures between 4 – 6 °C. L101–102 (now L106–108) were changed according to the referee's comment.

L106–108 (former L101–102) were changed to:
The rack is further equipped with a manually cooled rack section (ThermoSafe cooling packs at temperatures between -1 – -23 °C; Sonoco; Hartsville, South Carolina, USA) dedicated to the storage of liquids critical to hydroperoxide measurement and a constant pressure inlet (CPI) pump.

Line 106: You mention constant pressure here but don't mention a value until line 125 "...approximately ambient pressure..."— consider giving a value for the pressure control at line 106.

L111–113 (former L104) was changed according to the referee's recommendation.

L111–113 (former L104) was changed to:
Since the commercially available peroxide monitor was designed to perform measurements under standard conditions, the airflow system of HYPHOP was extended by the CPI system, ensuring constant pressure (approximately 1000 hPa) and airflow (2 slm) of the sampled air during airborne measurements at highly variable ambient pressure conditions.

Line 112: With a forward-facing inlet, how do particles and cloud droplets impact your measurement? Do you remove data affected by condensed phase material?

There are no known interferences between multiphase systems and the tracked gas phase species in the forward-facing inlet. Due to the high flow rate of the bypass, any potential impact of liquid and solid phase species is assumed to be minor.

Line 117: What airflow does the pump provide at maximum altitude (150 hPa ambient)?

The pump provides consistently an altitude independent airflow of approximately 10 slm. L115–117 (now L121–123) was changed according to the referee's comment.

L121–123 (former L115–117) was changed to:
The CPI system contains an internal CPI control unit and a Teflon-coated membrane pump (type MD 1C; Vacuubrand, Wertheim, Germany), generating a pressure altitude independent excess airflow of approximately 10 standard liters per minute (slm) to the sampling inlet of the monitor.

Line 126: I assume that the "reaction coil" mentioned here is the "stripping coil" shown in Figure 2. Making the body text match the figure text would be helpful.

L132–134 (former L126–128) was changed based on the referee's recommendation.

L132–134 (former L126 – 128) was changed to:
After traversing the inlet system, the ambient air is passed through a stripping coil (glass) with simultaneous injection of a precooled buffered sampling solution (4–6 °C; pH = 5.8–6.0) consisting of potassium hydrogen phthalate (KHP) and sodium hydroxide (NaOH; 1 M) in purified water.

Line 127 and elsewhere: You mention precooled solutions but nowhere in the manuscript do you give details on how the temperature control of the cooling is performed and how the temperatures of the solutions are monitored. Considering how important temperature is to your assumed sampling efficiencies, these details seem very important to the performance of your instrument and are critical to include in this manuscript.

The temperature control is performed manually by placing the analytical liquids in a sealed section of the rack filled with a sufficient amount (4 – 5) of ThermoSafe cooling packs at temperatures between -1 – -23 °C (Sonoco; Hartsville, South Carolina, USA). Unfortunately, due to weight restrictions no further temperature monitoring other than at the sampling coil is performed by the instrument. However, based on laboratory studies (measurement durations of 6 – 8 h) the temperature of the analytical liquids was determined to be at 4 – 8 °C at external temperature variations between 22 – 27 °C. Based on the reported relative cabin temperature variations of maximally 2 – 4 °C, the relative variation in the liquid temperature is assumed according to the laboratory tests to be at 4 – 6 °C. L106–108 (former L101–102) were changed according to the referee's recommendation as discussed above.

In Figures 3, 4, 5, and 7: consider changing the altitude unit from m to km to get rid of the "x103 ."

Altitude units in Fig. 3 – 5 & 7 were changed according to the referee's recommendation. Further, the altitude units were changed in the Supplement of the manuscript.

Lines 159-161: Calibrating with a liquid $H_2O_2$ standard calibrates the measurement to a certain extent, but it misses important stages of your instrument. Have you ever calibrated with gas phase peroxides? At multiple mixing ratios (or at least multiple $H_2O_2$ solution concentrations) to verify linearity?

Gas-phase calibrations of the instrument are performed using permeation sources described in detail in Sect. 2.2. (L220 ff.; former L203 ff.) in order to obtain the instrumental inlet transmission efficiency. Due to extended conditioning phases of the permeation sources, and restrictions in the laboratory equipment under field campaign conditions, additional frequent gas-phase calibrations are unfortunately not yet applicable during measurement flights. Based on previous research results on $H_2O_2$ sampling efficiency (99.8–99.9% at temperatures between 10 – 22 °C, Lee et al., 2000 and the references therein), no significant loss regarding the $H_2O_2$ sampling efficiency using gas-phase versus liquid-phase calibration technique is assumed. Thus, liquid calibrations are assumed to be sufficient in representing the absolute hydroperoxide mixing ratios. The instrumental linearity was confirmed using a series of freshly prepared liquid standards at concentrations between 0.0492–0.9842·$10^{-6}$ mol·$L^{-1}$. An exemplary linearity test is presented in the figure below. L159–161 (now L165–169) were extended regarding the referee's comment on instrumental linearity. Corresponding information was added to the Supplement of the manuscript.

[Figure]

**Figure S2: Hydrogen peroxide mixing ratios ([H2O2]) determined using HYPHOP plotted versus the theoretical hydrogen peroxide concentration (c(H2O2)$_{Theor}$) and the resulting linear regression (blue line).**

L159–161 (now L165–169) was changed to:
The calibration mode is performed on the ground before each measurement flight, during which a liquid $H_2O_2$ standard is measured to determine the hydroperoxide detection sensitivity and estimate a signal reference required to transfer the detected electric signal into hydroperoxide mixing ratios. Former tests using liquid calibration at various standard concentrations ($0.0492$–$0.9842 \cdot 10^{-6}$ mol·L$^{-1}$) confirm the instrumental linearity of the used calibration technique (Fig. S2 in the Supplement of this work).

Lines 162-163: Why is the HCl wash needed to precede the calibrations but not the ambient measurements?

HCl provides a valuable removal procedure for impurities and aged analytical liquids in-between the liquid changes performed approximately every 10 h. When the liquid containers are changed, automatically a new calibration is performed, which is then coupled with the HCl-flushing procedure. In order to reduce the time necessary to obtain a stable signal of the $H_2O_2$ liquid standard due to initial destruction of the $H_2O_2$ liquid standard by catalase residues in the reaction coil, and the tubing during the first step of the four-point calibration procedure, HCl is flushed prior to the first standard injection.

Lines 170-190 and related parts of the Supplement: All variables need to be defined and units given for each.

Variable definitions and corresponding units were added to the manuscript text and in the Supplement, as was recommended by the referee.

L176–198 (former L169–189) was changed to:
The concentration of hydrogen peroxide ($[H_2O_2]$; $[mol \cdot L^{-1}]$) in the liquid standard is determined by means of a redox titration with potassium permanganate ($KMnO_4$) under acidic conditions ($H_2SO_4$) as presented in the Supplement of this work (SR5–SR7 and Eq. S1). The corresponding mixing ratios can be calculated using the measured air and liquid flows of the instrument. Based on the four-point calibration process, the sensitivities of the consequent channels A and B ($s_A$, $s_B$; $[V \cdot ppbv^{-1}]$) are determined (Eq. 1–2).

$$s_A = \frac{U_{A,S} - U_{A,0}}{\mu_S} \qquad\qquad (1)$$

$$s_B = \frac{U_{B,S} - U_{B,0}}{\mu_S} \tag{2}$$

Here, $U_{A,S}$ [V] and $U_{B,S}$ [V] are the measured voltages during the first phase of the liquid calibration in the respective channels, and $U_{A,0}$ [V] and $U_{B,0}$ [V] are the corresponding signals measured in the second phase of the calibration. $\mu_S$ is the mixing ratio [ppbv] of the standard solution under consideration of the molar air volume ($V_m$; [$m^3 \cdot mol^{-1}$]) and the measured standardized mass flows ($Q_{Stripping}$ [L $\cdot$ min$^{-1}$], $Q_{Air}$ [slm]; Eq. S2 – S3):

$$\mu_S = \frac{[H_2O_2]}{10^4} \cdot 10^9 \cdot V_m \cdot \frac{Q_{Stripping}}{Q_{Air}} \tag{3}$$

The corresponding catalase destruction efficiency (typically between 0.95 – 0.99) of hydrogen peroxide in channel B, $\varepsilon$, is derived from the following equation:

$$\varepsilon = 1 - \left( \frac{U_{BK,S} - U_{BK,0}}{U_{B,S} - U_{B,0}} \cdot \frac{U_{A,S} - U_{A,0}}{U_{AK,S} - U_{AK,0}} \right) \tag{4}$$

with the voltages in both channels during the consecutive phases of the liquid calibration (liquid standard injection: $U_{A,S}$ [V], $U_{B,S}$ [V]; background measurement: $U_{A,0}$ [V], $U_{B,0}$ [V]; background measurement with catalase: $U_{AK,0}$ [V], $U_{BK,0}$ [V]; liquid standard injection with catalase: $U_{AK,S}$ [V], $U_{BK,S}$ [V]).

Based on the specific sensitivities, and the destruction efficiency, the absolute peroxide mixing ratios $\mu_{H_2O_2}$ [ppbv] and $\mu_{ROOH}$ [ppbv] in the sample are determined using the following equations:

$$\mu_{H_2O_2} = \frac{U_A - U_{AK,0}}{s_A} - \frac{(U_B - U_{BK,0})}{s_B} \cdot \varepsilon \tag{5}$$

$$\mu_{ROOH} = \frac{U_B - U_{BK,0}}{s_B} - (1 - \varepsilon) \cdot \mu_{H_2O_2} \tag{6}$$

Lines 180-184: What are typical values for the destruction efficiency?

Typical values for the destruction efficiency were added according to the referee's comment.

L188–193 (former L180–184) were changed to:
The corresponding catalase destruction efficiency (typically between 0.95 – 0.99) of hydrogen peroxide in channel B, $\varepsilon$, is derived from the following equation:

$$\varepsilon = 1 - \left( \frac{U_{BK,S} - U_{BK,0}}{U_{B,S} - U_{B,0}} \cdot \frac{U_{A,S} - U_{A,0}}{U_{AK,S} - U_{AK,0}} \right) \tag{4}$$

Line 190 through the end of Section 2.2: It would be good to introduce this section by naming the individual corrections and giving typical values (in pptv) for them.

L199–204 (former L190ff.) were changed based on the referee's recommendation.

L199–204 (former L190 ff.):
Due to the characteristics of the wet chemical measurement method, corrections regarding background signal variations (15–33 pptv between two consecutive background measurements and approximately 50–70 pptv over the duration of a typical measurement flight) and time modification due to the delay caused by liquid transport within the instrument (approximately 300 sec) have to be performed. Additionally, signal corrections regarding hydroperoxide transmission efficiencies due to potential wall losses at the inner surface of the sampling inlet (up to 300 pptv at 1.5 ppbv $H_2O_2$ and up to 100 pptv at 1.5 ppbv PAA, respectively), and sampling efficiencies have to be initially performed to obtain absolute hydroperoxide mixing ratios.

Lines 191-193: How much does the background typically vary, in terms of pptv of peroxide, between background measurements? Over the course of a flight?

The background varies typically between 15–33 pptv between single measurements and approximately by 50–70 pptv over the course of a flight, respectively. L199–209 (former L190–193) was changed based on the referee's comment.

L199–209 (former L190–193) was changed to:
Due to the characteristics of the wet chemical measurement method, corrections regarding background signal variations (15–33 pptv between two consecutive background measurements and approximately 50–70 pptv over the duration of a typical measurement flight) and time modification due to the delay caused by liquid transport within the instrument (approximately 300 sec) have to be performed. Additionally, signal corrections regarding hydroperoxide transmission efficiencies due to potential wall losses at the inner surface of the sampling inlet (up to 300 pptv at 1.5 ppbv $H_2O_2$ and up to 100 pptv at 1.5 ppbv PAA, respectively), and sampling efficiencies have to be initially performed to obtain absolute hydroperoxide mixing ratios.
In order to account for potential measurement divergencies and background signal alterations initiated by pressure, and temperature instabilities during the measurement flights, the frequently measured background (typically at least 3–4 times per flight) signal is interpolated according to the background measurement signals ($U_{A,0}$; $U_{B,0}$) obtained during the four-point calibration procedure. A typical background sampling frequency and duration are presented based on an exemplary measurement flight during the most recent research campaign, CAFE-Brazil in the Supplement of this work (Fig. S3).

Lines 199-200: Do you expect the inlet transmission efficiency to be temperature and pressure invariant?

Due to the low variability of the CPI pump's temperature (max. 3 °C), and thus inlet temperature, no temperature dependent variations of the inlet transmission are expected to occur. Analogously, as described in Sect. 2.1. (L117–131; formerly L111–125) nearly constant pressure and air flow are expected in the inlet system leading to the stripping coil. Thus, the inlet transmission efficiency is assumed to be pressure invariant as well. Additionally, Teflon-coating, and PFA tubing are assumed to display nearly constant chemical and physical behavior at temperatures up to 260 °C and no pressure instabilities in the range between 150–1000 hPa are reported in the literature, respectively (Hougham, 1999; Resnick and Buck, 1999).

Line 200: Regarding flow rates, 30 slm and 10 slm at all altitudes or is there a range of flow rates?

We apologize for the confusion. The flow rates of 30 slm at the bypass, and 10 slm at the inlet attached to the CPI pump are constant at all pressure altitudes. L217–219 (former L200–201) were improved in their readability.

L217–219 (former L200–201) were changed to:
Due to the high flow rates through the installed bypass (30 slm) and attached inlet (10 slm) constant at all pressure altitudes, potential wall losses in the tubing are assumed to have a minor impact on the hydroperoxide measurements.

Line 205: Using MHP instead of PAA for your inlet and CPI pump transmission tests would seem preferred, since MHP is your assumed organic peroxide.

We thank the referee for his comment. That is correct. However, MHP standards are not commercially available and their synthetical route is highly difficult under restricted campaign conditions. Further, as elaborated on in L223–226 (now L241–246), based on previous research (Fels and Junkermann, 1994; Lee et al., 1997; Valverde-Canossa et al., 2005; Hua et al., 2008), the composition of organic hydroperoxides in the BL as well as in air masses impacted for instance by biomass burning or in regions

with dense vegetation, might shift towards other organic hydroperoxide (as for instance PAA or ISOPOOH) as the most abundant trace species.

Line 221: Is the temperature of the sampling solution measured just prior to the reaction coil? Have you measured these sampling efficiencies with your instrument? And what would the sampling efficiencies be for PAA or HMHP?

The temperature of the sampling solution is measured at the sampling coil. Since the majority of the hydroperoxide synthesis routes require intensive care and caution, which cannot be realized without considerable effort, the instrumental sampling efficiencies are based on Lee et al. and the references therein (2000). Based on the mentioned literature, the sampling efficiencies for PAA and HMHP range between 0.82 and 1.00 within the temperature range of 10–22 °C and gas and aqueous flow rates of 2 slm and 0.004 L·min$^{-1}$, respectively, which compares well to the temperature and mass flow conditions of the HYPHOP monitor.

Lines 228-229: The transmission efficiency values would fit better at the end of the previous paragraph. Are these for the inlet + CPI pump or without the pump? It would be valuable to state both sets of values.

L228–229 (now L233–234) was changed according to the referee's recommendation.
As discussed in L217 ff. (former L200 ff.), due to high flow rates in the bypass and inlet, nearly no loss is assumed in the system without the CPI pump. Thus, the inlet transmission efficiency is determined as a ratio between the measured hydroperoxide mixing ratios with the CPI pump incorporated into the inlet system relative to the values measured without the CPI pump (assumed to be at 100% transmission efficiency) at a constant hydroperoxide mixing ratio provided by the permeation source.

L233 (former L216) was changed to:
The transmission efficiency during the most recent field campaign was calculated to be 82% (± 1.6%) for H$_2$O$_2$ and 95% (±1.1%) for ROOH.

Lines 239-241: have you experimentally confirmed that these additions do indeed eliminate all of the interferences?

Based on the expected amount of the discussed trace gases (at ppbv levels) leading to chemical interferences in the HYPHOP monitor and the reaction stoichiometries, concentrations of 0.068·10$^{-3}$ mol·L$^{-1}$ EDTA and 2.008·10$^{-3}$ mol·L$^{-1}$ HCHO translating into 0.134 mol·min$^{-1}$ and 3.953 mol·min$^{-1}$ of EDTA and HCHO, respectively (at stripping flow rate of 0.508·10$^{-3}$ L·min$^{-1}$) are assumed to be sufficient in cancelling the mentioned negative interferences.

Lines 253-254: Delete "...but also cabin..." and replace with "and"

L270–271 (former L253–254) were change based on the referee's recommendation.

L270–271 (former L253–254) was changed to:
Furthermore, cabin temperature and pressure may have a significant effect on signal detection in both channels.

Lines 255-263: This test is valuable but really only evaluates changes in background, not changes in instrument sensitivity, since no calibration standard was added. Are the flight maneuver effects expected to only affect background levels, not instrument sensitivity?

That is correct. The test evaluates the changes in the measurement background and thus, its performance during the subsequent flight measurements. Unfortunately, due to temporal instability of the liquid standard, calibration standard injections over the course of a test flight were impossible to perform in order to gain information on potential sensitivity changes initiated by different flight patterns. However,

as was discussed throughout Sect. 2.3. and exemplary shown in the Supplement of the manuscript (Figures S7–S8; former Fig. S5–S6), no significant performance instabilities, indicating reduction of the measurement sensitivities, were observed during measurement flights.

Line 275-280: 5 ppbv does not seem to be the appropriate reference here—shouldn't it be the signal levels of typical ambient mixing ratios since that's where the temperature variations are relevant (not during calibrations)? Using 5 ppbv badly underestimates the uncertainty caused by temperature-dependent noise, unless I'm missing something.

We apologize for the confusion and agree with the referee. L275–280 were removed from the manuscript.

Lines 323-325: Are the valves mounted vertically or horizontally? Some researchers have found improved immunity to aircraft motion by mounting valves horizontally.

The valves are mounted vertically. We thank the referee for her/his valuable comment and we would like to extend the discussion of future optimization approaches in Sect. 4. by the referee's suggestion with a citation note on the communication (private or namely). L483–486 (now L498–503) was changed according to the referee's comment.

L483–486 (now L498–503) was changed to:
However, instrument performance can be further optimized by minimizing the technical disturbances during the measurement flights. Potential higher temperature-dependent signal drifts are corrected by frequent background examination on the consecutive flight legs during the measurements. Furthermore, the instrument time resolution can be increased by reducing the length of the liquid supply tubing. Rapid line pressure inconsistencies might be avoided by mounting the valves horizontally instead of vertically (communication with anonymous referee, 2023/ [namely] communication, 2023).

Line 335: Change "...mixing ratio, which can be..." to "...mixing ratio that can be..."

L335–336 (now L350–351) was changed according to the referee's recommendation.

L335–336 (now L350–351) was changed to:
Applied to our measurement method, the limit of detection is defined as the hydroperoxide mixing ratio that can be distinguished from the background with a certainty of at least 95%.

Lines 249-352: As mentioned earlier, reproducibility of calibrations is more appropriately considered as part of instrument accuracy rather than precision. Precision itself will presumably be a function of mixing ratio and would be best quoted at a value much closer to ambient mixing ratios than the calibration standard value. Regarding accuracy and total measurement uncertainty, how well do you know the absolute value of your calibration standard?

We thank the referee for her/his comment. We refer to the discussion above, as elaborated with respect to the referee's comments 2 & 3. The absolute value of the calibration standard is given with a certainty of 95%.

Line 468: Considering the significant ozone interference and assumption of zero $H_2O_2$ in the LS to determine the interference value, a qualifier might be appropriate for the claim of measurements into the lower stratosphere.

L467 (now L482) was changed according to the referee's suggestion.

L467 (now L482) was changed to:
The presented wet chemical, dual-enzyme measurement method enables us to perform *in situ* measurements up to the tropopause and possibly lower stratosphere (1000 – 150 hPa).

**References**

Allen, H. M., Crounse, J. D., Kim, M. J., Teng, A. P., Ray, E. A., McKain, K., Sweeney, C., and Wennberg, P. O.: H 2 O 2 and CH 3 OOH (MHP) in the Remote Atmosphere: 1. Global Distribution and Regional Influences, JGR Atmospheres, 127, https://doi.org/10.1029/2021JD035701, 2022.

Crounse, J. D., McKinney, K. A., Kwan, A. J., and Wennberg, P. O.: Measurement of gas-phase hydroperoxides by chemical ionization mass spectrometry, Analytical chemistry, 78, 6726–6732, https://doi.org/10.1021/ac0604235, 2006.

Fels, M. and Junkermann, W.: The occurrence of organic peroxides in air at a mountain site, Geophys. Res. Lett., 21, 341–344, https://doi.org/10.1029/93GL01892, 1994.

Hougham, G. (Ed.): Fluoropolymers 2: Properties /  edited by Gareth Hougham … [et al.], Topics in Applied Chemistry, Kluwer Academic/Plenum, New York, London, 1999.

Hua, W., Chen, Z. M., Jie, C. Y., Kondo, Y., Hofzumahaus, A., Takegawa, N., Chang, C. C., Lu, K. D., Miyazaki, Y., Kita, K., Wang, H. L., Zhang, Y. H., and Hu, M.: Atmospheric hydrogen peroxide and organic hydroperoxides during PRIDE-PRD'06, China: their concentration, formation mechanism and contribution to secondary aerosols, Atmos. Chem. Phys., 8, 6755–6773, https://doi.org/10.5194/acp-8-6755-2008, 2008.

Kleindienst, T. E., Shepson, P. B., Hodges, D. N., Nero, C. M., Arnts, R. R., Dasgupta, P. K., Hwang, H., Kok, G. L., Lind, J. A., Lazrus, A. L., Mackay, G. I., Mayne, L. K., and Schiff, H. I.: Comparison of techniques for measurement of ambient levels of hydrogen peroxide, Environmental science & technology, 22, 53–61, https://doi.org/10.1021/es00166a005, 1988.

Lee, M., Heikes, B. G., and O'Sullivan, D. W.: Hydrogen peroxide and organic hydroperoxide in the troposphere: a review, Atmospheric Environment, 34, 3475–3494, https://doi.org/10.1016/S1352-2310(99)00432-X, available at: https://www.sciencedirect.com/science/article/pii/S135223109900432X, 2000.

Lee, M., Heikes, B. G., Jacob, D. J., Sachse, G., and Anderson, B.: Hydrogen peroxide, organic hydroperoxide, and formaldehyde as primary pollutants from biomass burning, J. Geophys. Res., 102, 1301–1309, https://doi.org/10.1029/96JD01709, 1997.

Mackay, G. I., Mayne, L. K., and Schiff, H. I.: Measurements of H 2 O 2 and HCHO by Tunable Diode Laser Absorption Spectroscopy During the 1986 Carbonaceous Species Methods Comparison Study in Glendora, California, Aerosol Science and Technology, 12, 56–63, https://doi.org/10.1080/02786829008959325, 1990.

Resnick, P. R. and Buck, W. H.: Teflon® AF: A Family of Amorphous Fluoropolymers with Extraordinary Properties, in: Fluoropolymers 2: Properties /  edited by Gareth Hougham … [et al.], edited by: Hougham, G., Kluwer Academic/Plenum, New York, London, 25–33, https://doi.org/10.1007/0-306-46919-7_2, 1999.

St Clair, J. M., McCabe, D. C., Crounse, J. D., Steiner, U., and Wennberg, P. O.: Chemical ionization tandem mass spectrometer for the in situ measurement of methyl hydrogen peroxide, The Review of scientific instruments, 81, 94102, https://doi.org/10.1063/1.3480552, 2010.

Staffelbach, T. A., Kok, G. L., Heikes, B. G., McCully, B., Mackay, G. I., Karecki, D. R., and Schiff, H. I.: Comparison of hydroperoxide measurements made during the Mauna Loa Observatory Photochemistry Experiment 2, J. Geophys. Res., 101, 14729–14739, https://doi.org/10.1029/95JD02197, 1996.

Valverde-Canossa, J., Wieprecht, W., Acker, K., and Moortgat, G. K.: H2O2 and organic peroxide measurements in an orographic cloud: The FEBUKO experiment, Atmospheric Environment, 39, 4279–4290, https://doi.org/10.1016/j.atmosenv.2005.02.040, 2005.